# Phytochemical and Biological Evaluation of a Newly Designed Nutraceutical Self-Nanoemulsifying Self-Nanosuspension for Protection and Treatment of Cisplatin Induced Testicular Toxicity in Male Rats

**DOI:** 10.3390/molecules26020408

**Published:** 2021-01-14

**Authors:** Sherif R. Abdel-All, Zeinab T. Abdel Shakour, Dalia M. N. Abouhussein, Enji Reda, Thoraya F. Sallam, Hala M. El-Hefnawy, Azza R. Abdel-Monem

**Affiliations:** 1Phytochemistry and Natural Product Department, Egyptian Drug Authority, Giza 12553, Egypt; sherif.EDA2020@gmail.com; 2Pharmaceutics Department, Egyptian Drug Authority, Giza 12553, Egypt; dalia.abouhussein@edaegypt.gov.eg or; 3Pharmacology and Toxicology Department, Faculty of Pharmacy, Sinai University, East Kantara Branch, New City, El Ismailia 41611, Egypt; Enji_pharma81@hotmail.com; 4Histology and Cytology Department, Faculty of Veterinary Medicine, Cairo University, Giza 12211, Egypt; thoraya.vet2020@hotmail.com; 5Pharmacognosy Department, Faculty of Pharmacy, Cairo University, Cairo 11562, Egypt; hala.elhefnawy@pharma.cu.edu.eg (H.M.E.-H.); azza.abdelmonem@pharma.cu.edu.eg (A.R.A.-M.)

**Keywords:** new nutraceutical formulation, *Spirulina platensis*, *Tribulus terrestris* L., fish oil, SNESNS, cisplatin, histopathology, biochemical markers

## Abstract

The incorporation of cisplatin (CP) as a cytotoxic antineoplastic agent in most chemotherapeutic protocols is a challenge due to its toxic effect on testicular tissues. Natural compounds present a promising trend in research, so a new nutraceutical formulation (NCF) was designed to diminish CP spermatotoxicity. A combination of three nutraceutical materials, 250 mg *Spirulina platensis* powder (SP), 25 mg *Tribulus terrestris* L. extract (TT), and 100 mg fish oil (FO) were formulated in self-nanoemulsifying self-nanosuspension (SNESNS). SP was loaded into the optimized self-nanoemulsifying system (30% FO, 50% span 80/cremophor EL and 20% isopropanol) and mixed with TT aqueous solution to form SNESNS. For the SP, phytochemical profiling revealed the presence of valuable amounts of fatty acids (FAs), amino acids, flavonoids, polyphenols, vitamins, and minerals. Transmission electron microscopy (TEM) and particle size analysis confirmed the formation of nanoemulsion-based nanosuspension upon dilution. Method validation of the phytochemical constituents in NCF has been developed. Furthermore, NCF was biologically evaluated on male Wistar rats and revealed the improvement of spermatozoa, histopathological features, and biochemical markers over the CP and each ingredient group. Our findings suggest the potential of NCF with SNESNS as a delivery system against CP-induced testicular toxicity in male rats.

## 1. Introduction

Cisplatin (CP) is a main chemotherapeutic treatment that has been used against different types of solid malignant tumors, i.e., testicular germ cell tumor (TGCT) [1]. CP was reported to destroy cells through different mechanisms including DNA impairment, reactive oxygen species (ROS) construction, and apoptosis encouragement as a DNA alkylating agent [2]. Acute exposure to CP resulted in strong destruction of the seminiferous epithelium and this damage could be extended to the germ cells causing inhibition of its proliferation, meiosis, and differentiation to spermatozoa. The cellular mechanism by which CP-induced testicular toxicity involves disruptions of the redox equilibrium of testicular tissues via elevation of mitochondrial impermanent, oxidative/nitrosative stress, and lipid peroxidation resulting in protein synthesis retardation and DNA destruction [3]. Interestingly, growing evidence revealed that combination therapy of CP with antioxidants could be valuable to overwhelm this special reproductive toxicity. Natural products have always been a venue for the search for new drugs or leads. Spirulina (SP), *Arthrospira platensis*, (family: *Oscillatoriaceae*), is cyanobacteria (blue-green microalgae) used conventionally as an antioxidant source against ROS. The Food and Drug Administration (FDA) suggested that SP was a safe food supplement in 2012 [4]. SP was reported to be a rich natural source of proteins, carbohydrates, polyunsaturated fatty acids (PUFA), sterols, zinc, manganese, iron, calcium, magnesium, selenium, vitamin B_12_, E, C, tocopherols, carotenoids, and xanthophyll phytopigments [5]. Additionally, SP has chemoprotective, anti-inflammatory, and antioxidant effects. The phycocyanin pigment present in SP was reported to have a defensive effect on DNA damage induced by CP and can inhibit oxidative stress with powerful antioxidant activity [6]. *Tribulus Terrestris* L. (TT), family *Zygophyllaceae,* is a shrub, growing in subtropical climates all over the world [7]. It consists of numerous biologically active constituents like steroidal saponins, unsaturated fatty acids, alkaloids, flavonoids, tannins, and vitamins. The main active component of TT is protodioscin, a steroidal glycoside, has recovered the levels of testosterone, dihydroepiandrosterone and dihydrotestosterone and thereby improving erectile dysfunction, libido, and low seminological indices [8]. TT was found to induce powerful protection against cadmium-induced testicular toxicity through retardation of testicular peroxidation by antioxidant and metal chelating activity and also, maybe indirectly by stimulating the production of testosterone from Leydig cells [9]. Eicosapentaenoic acid (EPA), docosahexaenoic acid (DHA), and alpha-linolenic acid are the main fatty acids (FAs) in fish oil (FO) [10]. PUFA have played an important role in male fertility through the improvement of fluidity to the plasma membrane of sperm for the unification actions that characterize fertilization and noteworthy constituents of a specific class of fucosylated glycosphingolipids are crucial for male fertility [11]. No studies have been performed on the use of the combination of these three constituents. This stimulated our consideration to investigate the potential protective and/or recovery of TT, FO, and SP extracts separately and simultaneously in a novel nano-sized formulation for CP-induced testicular toxicity in male rats. Nanoemulsion and self-nanoemulsifying drug delivery systems (SNEDDS) are drug delivery systems that have been proved to enhance the solubility and hence the absorption and the bioavailability of poorly water-soluble drugs. These systems assisted the drug solubilization in the intestinal fluids, when ingested, with the consequent enhancement of drug absorption particularly in the presence of surfactants and co-surfactants [12]. The use of SNEDDS has been challenged by the limited solubility of large doses of some drugs as in the case of SP extract which necessitates increasing the drug content in the formulation. Self-nanoemulsifying self-nanosuspension (SNESNS) is associated with the benefits of both SNEDDS and nanosuspension. The drugs are incorporated in the isotropic mixture of oil, surfactant and co-surfactants as both solubilized molecules and suspended microparticles. Once the formulation is exposed to the gastrointestinal fluids, it is spontaneously emulsified developing a nanoemulsion-based nanosuspension (NENS) of the drug combination leading to quick drug dissolution from the surface of the suspended microparticles that enhances both drug permeation and oral absorption [13,14,15]. This study aims to formulate an NCF in a SNESNS design in which each gram contains 250 mg SP, 100 mg FO, and 25 mg TT as active forms. The new formulation was fully characterized physically, chemically with complete validation of the used methods for the assay of each ingredient. Moreover, an in vivo study was performed by treating separate groups of male rats with each ingredient and the selected formula for 10 days then injecting them with CP, then completing the treatment for one week. Several parameters have been evaluated for biological investigation, including spermatological changes, testicular histopathological alterations, hormonal changes, oxidative stress parameters, inflammatory markers, and apoptotic status.

## 2. Results and Discussion

### 2.1. Phytochemical Investigation of Spirulina Powder

This study focused on the phytochemical investigation of SP commercialized in the Egyptian markets as a raw powder sold in dietary stores and it is not manufactured as a pharmaceutical dosage form. The study includes the analysis of amino acids, FAs (omega FAs), carotenes, vitamins, and minerals.

#### 2.1.1. GC/MS Analysis of Lipid Content

There were 15 unsaponifiable matters (USM) (hydrocarbons) identified, representing 65.74% of the lipoidal content. Heptadecane was the main hydrocarbon identified (32.26%) followed by squalene (8.1%) as presented in Figure 1A. Seven fatty acid methyl esters (FAME) were identified representing 71.38% of the saponifiable matter. Hexadecanoic acid (palmitic acid) was the main FA identified (25.53%) followed by 9,12-octadecadienoic acid (linoleic acid; LA) (15.08%) and γ-linolenic acid (GLA) (14.52% area) as presented in Figure 1B. SP is considered an important source of omega FAs as GLA, LA, and oleic acids [16]. Myristic, heptadecanoic, stearic, oleic, palmitoleic, omega-3, omega-6, LA, GLA, and palmitic acid, the highest saturated FA, were also identified in SP. These results are in accordance with the ones in [17].

#### 2.1.2. The Total Protein and Amino Acids Composition

The total protein content represented 68.77% of SP, which was found to be in accordance with the labeled amount of the total protein (65–75% protein content) in the marketed product. The essential amino acids represented 28.7% while non-essential amino acids represented 40% from SP. Glutamic acid is the main non-essential amino acid (14.1%) followed by aspartic acid (8.52%) while leucine is the main essential amino acid represent (6.4%) followed by valine (5.3%) as presented in Figure 2. Previously, the total protein content was reported to be 56.79% [18] and other studies reported it to be 62.84% [19], which was slightly below the results of this study. Other studies reported leucine as the main essential amino acid with a percentage of 5.5% and glutamic acid as the main non-essential amino acid with a percentage of 9.2% [17], which was in accordance with the results of this study.

#### 2.1.3. Determination of Total Phenolics (TPC) and Total Flavonoids (TFC) Contents

The TPC content was (58.64 mg gallic acid equivalent (GAE)/g dry SP). It was reported that aqueous extract of SP exhibited the highest total phenolic content (43.2 mg GAE/g ext.) than 100% methanol extract (24.4 mg GAE/g ext.) [20] and (33.57 mg GAE/g D. Wt.) [21], which were found to be slightly less than the results of this study. While, the TFC content was (0.864 mg rutin equivalent (RE)/g dry SP). It was reported that SP contains total flavonoids (0.166 mg quercetin equivalent (QE)/g dry extract (DE) [22] and (7.11 mg QE/g DW) [18]. For the first time, TFC was estimated in commercial SP in Egypt using rutin as a reference standard.

#### 2.1.4. HPLC Determination of Vitamin B_12_, Vitamin C, and Vitamin E

SP contained 3.2 mg/100 g of cyanocobalamin (vitamin B_12_) and 191 mg/100 g of ascorbic acid (vitamin C) while α-tocopherol (vitamin E) was undetectable. It was reported that 175 μg/100 g of vitamin B_12_ and 9.86 mg/100 g of vitamin E were detected in SP [19]. Other studies reported 162 μg/100 g of vitamin B_12_ [23] and 60 mg/100 g vitamin E in SP [18] However, these results were inconsistent with the results of this study as the content of vitamin B12 was higher than reported in the previous studies and vitamin E was undetectable as shown in Appendix A.

#### 2.1.5. Determination of Mineral Content

SP contained 170 mg/100 g of iron, 160 mg/100 g of zinc and 20 mg/100 g of copper. It was reported that 273.197 mg/100 g of iron, 1.2154 mg/100 g of copper, and 3.6229 mg/100 g of zinc were detected in SP [19]. Other studies reported 87.4 mg/100 g of iron, 1.45 mg/100 g of zinc, and 0.47 mg/100 g of copper in SP [23] and these results were inconsistent with the results of this study as shown in Appendix A. 

From the previous results, the nutritional value of the SP is summarized in Table 1.

### 2.2. Formulation of a New Nutraceutical SNESNS

#### 2.2.1. Screening of SNEDDS Components

The reduced solubility of various herbal extracts is a real challenge in developing aqueous oral formulations. SNEDDS is an ideal system for a combination dosage form comprising oil and solid herbal extracts. Moreover, SNESNS is an innovative drug delivery system that can deliver drugs of large dose and limited solubility. The solubility of FO in different surfactants and co-surfactants was investigated. It showed the highest solubility in span 80/cremophor EL as a surfactant and isopropyl alcohol as a co-surfactant. 

#### 2.2.2. Construction of Ternary Phase Diagram

A ternary phase diagram was designed to distinguish the points that can develop SNEDDS. About the results obtained from the solubility of FO in different surfactants and co-surfactants, span 80/cremophor EL and isopropyl alcohol were chosen as the surfactant and co-surfactant, respectively as presented in Appendix A, Figure 3 shows the constructed phase diagrams. F1-F6, prepared using a high concentration of FO (40, 50%), gave a turbid liquid with the appearance of oil globules indicating the instability of the system. On the other hand, all the developed formulations (F7–F12) prepared using 30, 35% FO displayed a clear liquid. 

#### 2.2.3. Characterization of SNEDDS

##### Determination of Emulsification Time (ET), Dispersibility, and %Transmittance (%T)

The successful selection of the surfactant and co-surfactant leads to the effective emulsification of the developed system. The surfactant/co-surfactant system represents the key factor in SNEDDSs by hindering the globule coalescence through decreasing the globule surface tension and consequently, enhancing the system stability [24]. Surfactant hydrophile–lipophile balance (HLB) value has a great effect on the emulsification process and the globule dimension. Span 80 has an HLB value of 4.3 while cremophor EL has HLB between 12–14. Mixing Span 80 with cremophor EL may increase the total surfactant HLB which may enhance the developed nanoemulsion stability as higher HLB lead to more stable nanoemulsion upon contact with aqueous media [25]. F10 and F11 displayed the shortest ET of 10 ± 1.2 s and 12 ± 1.5 s respectively followed by F12 (20 ± 1.5 s); showing grade A nanoemulsions upon dilution. On the other hand, F7–F9 showed a slower ET of 36 ± 2.3 s, 40 ± 2.1 s, and 45 ± 2.5 s respectively, and formed a faintly translucent (grade B) nanoemulsion. F1–F6 showed milky emulsions that represented grade C nanoemulsions. F10 and F11 showed a high % transmittance 91.5 ± 2.3% and 92.3 ± 1.6% respectively. Other formulations showed a lower % transmittance.

##### Simulation of the Physiological Dilution

It is essential to guarantee that the nanoemulsion does not show any physical instability after different dilutions in any gastrointestinal fluid. F11 retained its physical stability for 24 h. In contrast, turbidity occurred when F10 was 50-fold diluted with distilled water, 0.1 N HCl, and phosphate buffer (pH 7.4) and slight turbidity when diluted to 100 and 1000-fold using the same media while F12 exhibited a turbid appearance when diluted to 50 and 100-fold in case of the three investigated media and a slightly turbid when diluted to 1000-fold using the same media. Therefore, F10 and F12 were eliminated from further characterization.

##### Globule Size(GS) Analysis, Polydispersity Index (PDI), and Zeta Potential

The average GS of the examined F11 SNEDDSs after 50-folds dilution was 43.19 nm as depicted in Figure 3. The PDI was 0.55. SNEDDS represents a formulation that produces nanoemulsion of GS below 100 nm [26]. Accordingly, F11 can be considered as SNEDDS. Moreover, the formulation showed a zeta potential of −17.8. In comparatively small particle size globules, dispersions display a high zeta potential that allows the globules to repel due to the repulsion of the similarly charged particles leading to nanoemulsion stability [27].

##### Preparation of the Extract Loaded SNESNS

Extract loaded SNESNS was prepared to optimize SP content in the formulation by homogenization of SP in the selected SNEDDS to comprise both solubilized and suspended SP molecules to finally achieve a consistent green dispersion. Afterward, this dispersion was mixed with TT solution where the SNESNS was emulsified to form o/w nanoemulsion based nanosuspension which gave an opalescent green solution upon dilution 1:200 with distilled water [28].

#### 2.2.4. Characterization of the Extract Loaded SNESNS

##### Particle Size Analysis (PS), Polydispersity Index (PDI), and Zeta Potential

As presented in Figure 4, the tested SNESNS showed two peaks. One of higher intensity with an average PS of 332.2 nm represents the formed nanoparticles, while the other peak of lower intensity showed a mean PS of 45.6 nm, which corresponds to the nanoemulsion droplets. The PDI of the SNESNS was 0.59. The zeta potential of the prepared SNESNS was −19.5 which indicates a nanosuspension of good stability that can withstand the aggregation and precipitation of particles.

##### Transmission Electron Microscopy (TEM)

The morphology of a nanosuspension is a vital parameter to be studied. The TEM micrographs in Figure 5C. showed that the nanoemulsion globules were formed after diluting the SNEDDS. These globules are white spherical globules with a homogenous nano-sized distribution. This may be clarified by the water solubility of the used phosphotungstic acid that tends to surround the nanoemulsion globules while the solvents evaporate and hence the globules appear brighter against the dark surroundings [29]. Moreover, stirring SNESNS in distilled water resulted in spontaneous nano-sizing of the suspended SP in SNESNS and the formation of globular nanoparticles, as shown in Figure 5A,B [28].

### 2.3. Methods Validation for NCF Quantitative Analysis

#### 2.3.1. Optimization of Chromatographic Conditions

The developed methods were validated in terms of linearity, accuracy (%recovery), precision, and sensitivity (LOD, LOQ) according to ICH guidelines, [30,31,32].

#### 2.3.2. Method Validation for GC/FID Assay of FAME in NCF and Its Ingredients

##### Content of Palmitic Acid, GLA, LA, EPA, and DHA

The calculated content of palmitic acid, GLA, and LA in SP was 132 mg/g pd., 58.74 mg/g pd., and 27.7 mg/g pd. respectively. TT was found to contain 21.2 mg/g palmitic acid and 103.2 mg/g LA. FO was found to contain 69.37 mg/g oil palmitic acid, 183.73 mg/g LA, 237.95 mg/g eicosapentaenoic acid (EPA), and 199.9 mg/g docosahexaenoic acid (DHA). The NCF was found to contain 110.68 mg/g palmitic acid, 16.8675 mg/g GLA, 104.12 mg/g LA, 22.04 mg/g eicosapentaenoic acid (EPA) and 18.58 mg/g form docosahexaenoic acid, these results were shown in Figure 6, Appendix A.

##### Validation of the GC/FID Method

The standard calibration curves were constructed for hexane stock solutions with appropriate concentrations of palmitic acid, GLA, LA, EPA, and DHA. Each analyte in four concentrations was injected three times, and the calibration curves were created by plotting the peak areas under the curve versus the amount of the analytes. The standard calibration curves of palmitic acid, GLA, LA, EPA, and DHA are expressed by the equations as y = 326.43x − 1.7899, R^2^ = 0.9991 and a slope equals 326.43, y = 357.46x − 6.8045, R2 = 0.9999 and a slope equals 357.46, y = 1252.5x − 231.64, R2 = 0.9923 and a slope equals 1252.5, y = 2509.7x − 9.2343, R2 = 0.9817 and a slope equals 2509.7, y = 1909.9x + 151.15, R2 = 0.9912 and a slope equals 1909.9, respectively and shown in Appendix A.

The intra-day and inter-day precision of the method technique was determined by analyzing three replicates (*n* = 3) of the SP, TT, FO, and NCF in one day (intra-day precision) and three days interval (inter-day precision) for palmitic acid, GLA, LA, EPA, and DHA. The precision was expressed in %RSD and it was found to be less than 2 for intra-day which was within the acceptable range and more than 2 for inter-day analyses indicating that the samples should be freshly prepared before injection and not be stored even in a refrigerator. The accuracy was assessed at four concentrations each of palmitic acid, GLA, LA, EPA, and DHA which were performed in the successive analysis (*n* = 3) using the planned method, and the value was expressed as a percentage of recovery which were found to be 99.88%, 99.88%, 97.05%, 101.45%, and 99.415%, respectively as shown in Appendix A. All experimental results were found to be in the range of the acceptability for accuracy suggesting that this method is sensitive and accurate for the determination of each constituent alone and together in a sample.

#### 2.3.3. Method Validation for HPLC and HPTLC Assay of β-Carotene (Vitamin A) in SP and NCF

##### Assay of β-Carotene (Vitamin A) by HPLC and HPTLC

The calculated content of β-carotene in SP and NCF by HPLC is (0.44 mg/g pd. and 0.17 mg/g formula), respectively. The calculated content of β-carotene in SP and NCF by HPTLC is (1.63 mg/g pd.) for SP, while β-carotene was not separated from the NCF by HPTLC. These results are shown in Figure 7 and Figure 8.

##### HPLC and HPTLC Method Validation for Assay of β-Carotene in SP and NCF

Linearity was determined with calibration curves, which were created by plotting the measurements of area peak versus the concentration (mg/mL) of β-carotene. Four concentrations of each analyte were injected in triplicate. The standard calibration curve of β-carotene obtained from HPLC and HPTLC is a linear straight line expressed by the equation y = 54,020x − 9.6388, R^2^ = 0.9995 and a slope equals 54,020 and y = 42,777x + 6989.3, R^2^ = 0.9814 and a slope equals 42,777, respectively. These results are shown in Appendix A.

The intra-day precision for the analytical method of β-carotene in SP and NCF expressed as %RSD either by HPLC and HPTLC were found to be less than 2 which was within the acceptable range of the precision while inter-day precision obtained either by HPLC and HPTLC were found to be more than 2 suggesting that the samples should be freshly prepared prior injection and should not be stored even in a refrigerator, also light protected glass should be used during the preparation of samples. Carotenoids were characterized by the conjugated polyene chain that makes these compounds subject to destruction from several agents. The terminal end groups of carotenoids were highly liable for deterioration from certain environments. Also, carotenoids can react easily in atmospheric oxygen especially when present as a purified form in organic solvents [33]. The accuracy was evaluated at four different concentrations of β-carotene by HPLC and HPTLC which was expressed as percentage recovery and found to be 100.2051% and 99.62%, respectively as shown in Appendix A. It was noticed that β-carotene was not separated and measured from NCF by the HPTLC method as presented in Figure 8, suggesting that the HPLC method was the most sensitive and accurate method for the quantification of β-carotene in the SP and NCF.

#### 2.3.4. Method Validation for UV Colorimetric Assay of Total Steroidal Saponins in TT and NCF

##### Content of Total Steroidal Saponins by UV Colorimetric Method

The total steroidal saponin content for TT and NCF was 609 mg/g ext., and 17.72 mg/g formula, respectively.

##### Validation of UV Colorimetric Method for Assay of Total Steroidal Saponins in TT and NCF

Linearity was evaluated through a standard calibration curve using TT working standard extract (45% steroidal saponins) by plotting the measurements of absorbance versus concentration (mg/mL) of total steroidal saponins. Four concentrations of each analyte were measured in triplicate. The standard calibration curve of steroidal saponins was expressed by the equation y = 0.0718x + 0.0034, with a slope equal to 0.0718 and R^2^ = 1, these results were shown in Appendix A.

The precision (repeatability) was performed for UV estimation of the total steroidal content method in two manners; the intra-day and inter-day precision expressed as %RSD which is found to be less than 2% for intra-day precision and more than 2% for inter-day precision. It is important to take into consideration that the samples should be freshly prepared before being measured on a UV spectrophotometer and not to be stored even in a refrigerator. For NCF, it was necessary to perform defatting using petroleum ether to remove any undesirable color for the chlorophyll of SP and FO and also to perform a blank using SP and FO without TT under the same conditions as the sample to mimic the conditions of the samples during extraction and avoid any calculation errors concerning the recovery of the sample. The accuracy was evaluated at four different concentrations for total steroidal saponins of TT by UV colorimetric method which was expressed as percentage recovery and found to be 99.97%. as presented in Appendix A. The chemical constituents of the NCF is summarized in the Table 2.

### 2.4. Biological Evaluation of the NCF

#### 2.4.1. Sperm Count and Motility

The sperm count and motility were observed clearly in CP intoxicated group compared to the normal control Appendix A in and Figure 9. CP showed a significant decline of sperm count (77%) and motility (50%) and life-dead% (42%), along with a two-fold increase of sperm abnormalities compared to normal control rats. Moreover, a significant increase and improvement of sperm count, motility, and morphology were observed in rats treated with cisplatin with NCF compared to other groups (4.33-fold from CP; 2.88-fold from FO; 1.3-fold from TT and 1.08-fold from SP). Conversely, TT, SP, and NCF significantly eliminated the changes of sperm-specific parameters, in comparison with the CP-untreated group. No important alteration was noticed in FO alone treated rats in the level of sperm abnormalities and motility when compared with the CP-untreated group.

#### 2.4.2. Histopathological Investigation

The histopathological investigation revealed that CP treated groups (+ve control group) showed significant damage to the spermatogonial cells and Sertoli cells with severe degeneration, vascular changes, and many records of spermatid giant cells as well as intra-luminar desquamation of germinal epithelial cells as shown in Figure 10B. Administration of FO to CP treated group revealed partial recovery with remarkable enhancement and improved damage in spermatogonial cell, Sertoli cells, and interstitial tissues with mild records of degenerative changes, as shown in Figure 10C. Treatment with TT could restore the normal structure of the seminiferous tubules with regular basement and apparent intact normal spermatogenic cells, Sertoli cells, interstitial tissue and vasculature as presented in Figure 10D. SP treated group showed a significant increase in the number of spermatids and spermatozoa in the lumen of the seminiferous tubules with minimal lesions as presented in Figure 10E. Administration of NCF with nano-designed form in comparison with each ingredient alone and with the reference drug, TT, showed well organized, intact, active, normal seminiferous tubules, showing a large number of spermatids and spermatozoa without histological lesions. These results suggested that the newly designed formula had more valuable and effective protection against CP-induced reproductive damages than the other treatments, indicating the beneficial role of it in the improvement, recovery, and restoration of spermatogenic cells, as shown in Figure 10F.

#### 2.4.3. In Vitro Antioxidant (DPPH Radical Scavenging Activity) for SP and NCF

The SP and NCF showed significant antioxidant activity expressed as % inhibition (25.4% and 35.3%) with IC_50_ (1.606 mg/mL and 1.122 mg/mL), respectively, the results are listed in Appendix A. The NCF showed higher antioxidant activity with a lower IC_50_ value (1.122 mg/mL) compared to gallic acid as a reference standard with IC_50_ (0.016 mg/mL). It was suggested that other ingredients in the NCF have additive antioxidant action than the SP alone. Previous studies reported that ethanol extract of SP exhibited the strongest antioxidant activity using the DPPH assay method at a concentration of 100 µg/ml with % inhibition 27.88% compared with ascorbic acid as reference standard with % inhibition 87.57% [34].

#### 2.4.4. Biochemical Analysis

##### Effect of Cisplatin-Induced Alterations in Serum Testosterone

The CP-treated group showed a marked decline in serum testosterone level to about 77% of the control group. Treatment with TT, SP, FO, and NCF completely alleviated the changes induced by CP to different levels, compared to the CP-untreated group. There was a significant change in NCF that showed a significant elevation for serum testosterone level when compared with other groups. Results are presented in Appendix A and Figure 11.

##### Changes in Testicular Oxidative Stress and Antioxidant Markers

CP markedly elevated the testicular content of (A) malondialdehyde (MDA) (6-fold) but abated that of (C) glutathione (GSH), (D) Total antioxidant capacity (TAC), and (E) Nuclear factor (erythroid-derived 2)-like 2 (Nrf2 by 80%, 60%, and 75%, respectively, as compared to normal control rats. Nevertheless, all treatments showed marked variations in oxidative stress and antioxidant markers to different levels with the NCF, which showed the best effect in signifying the suppression of oxidative stress and boosting of testicular antioxidant defenses, compared to the CP-untreated group which is presented in Appendix A and Figure 12.

##### Changes in Testicular Inflammatory Status

The testicular content of (A) nuclear factor kappa-light-chain-enhancer of activated B cells (NF-κB) and (B) interleukin-6 (IL-6) was notably elevated (6, 3.5-fold, respectively) in the CP-treated rats relative to the normal control as shown in Figure 13. The content of NF-κB was attenuated in all treated animals in ascending order of TT, SP, and FO compared to the CP-untreated group. However, FO, TT, and SP cannot significantly decrease the level of the content of IL-6, compared to the CP-untreated group. On the other hand, the greatest lowering effect on NF-κB and IL-6 was observed with the NCF by 70% and 50%, respectively, in relation to the insult, reaching the normal level, and the results are presented in Appendix A and Figure 13.

##### Effect on Testicular Caspase 3

CP amplified testicular caspase 3 by two-fold, compared with normal control, as shown in Figure 14. An effect that alleviated by management with TT (32%), FO (36%), compared to the CP-untreated group as shown in Appendix A and Figure 14. Moreover, CP treated with SP and NCF showed a noticeable decline in caspase 3, in relation to the insult, and caused restoration of the normal values.

### 2.5. Discussion

CP is a strong alkylating chemotherapeutic agent and its consumption in many chemotherapeutic protocols can lead to harmful side effects, including testicular toxicity through induction the release of reactive oxygen species (ROS) that causes a disequilibrium between the synthesis of oxidants and removal by the antioxidant defense mechanism [35]. The anti-inflammatory, antioxidative, and detoxification enzymes in addition to proteins that assist in the repair or removal of damaged macromolecules as PARP-1 was enhanced by a dominant controller Nrf2 [36]. The normal spermatogenesis along with remaining the normal arrangement of seminiferous tubules are regulated and maintained through the essential testosterone hormone [37]. The spermatogenic impairment in the CP treated rats indicated in the current study isn’t only the result of the reduced testosterone level, but it may be also due to the formation of ROS in the testicular tissues as they exert a detrimental effect on spermatogenesis [38]. In the current study, a significant testicular injury of CP was evidenced by the histopathological anomalies alongside spermatogenesis inhibition. In addition, a marked reduction of serum testosterone levels was also observed. The increased generation of free radicals induced by CP resulted in severe damages on testicular Leydig and Sertoli cells and thus remarkable reduction in the hormonal level as one of the possible mechanisms [39]. These findings were observed also in other studies confirming the cytotoxic effect of CP [40]. The present study was implemented to investigate the potential protective and/or recovery of NCF against CP induced testicular toxicity in male rats. This study has not been exploited before. Interestingly, treatment with the prepared NCF demonstrated significant efficiency for the reduction of testicular injury and disturbed spermatogenesis. These interpretations are in line with the growing evidence for TT, which found that the major steroidal frostanol saponin, protodioscin, encouraged spermatogenesis by stimulating the production of pituitary gonadotropins that triggered testosterone and improved sperm count by affecting Sertoli cells [41]. The presence of powerful antioxidant components like C-phycocyanin, phenolic contents, β-carotene, and vitamin C in SP were found to be mainly responsible for the antioxidant and anti-inflammatory activity through inhibition of Nf-ƙB, induction of Nrf2, maintaining the endogenous antioxidants and inhibiting the rise of testicular NO and TNF-α, thus decreasing oxidative stress and relieving the pathological changes leading to an improvement of testosterone level. Further, they have a dominant protective effect against cellular DNA and macromolecule damage and support the renovation and restoration of damaged cells. They also have a protective role against oxidative damage of P_450_ systems in Leydig cells [42]. The EPA of FO was found to have a master role in testosterone metabolism, combined into the testicular-interstitium cells plasma membrane in a distinctive way, also It was found to be arranged in Leydig cells with unknown manner [43]. Moreover, our results revealed that NCF co-treatment with CP significantly alleviated the reduction in serum testosterone concentration compared to the CP-untreated group for about 3.4-fold. The powerful efficacy of Omega-3 FA may be related to changes in the sperm phospholipid FA profile. Omega-3 FA was found to improve the characteristic flagellar motion of sperm and enhance the flexibility of the sperm membrane [44]. The production of antioxidant and cytoprotective enzymes such as SOD, catalase, GSH, haem-oxygenase-1 (HO-1) and NADPH quinone oxidoreductase (NQO1) and increased synthesis of GSH, NADPH and multidrug transporters was engendered through Nrf2 activation mediated by CP induced reactive oxygen species [45]. MDA production was increased as a result of successive depletion of cellular antioxidant defenses, as antioxidant enzymes, and reduced glutathione leads to triggered lipid peroxidation of mitochondrial and sperm membranes [46]. The greater rates of metabolism and replication of the testicular tissue rendering it to be more subjected to oxidative damage by ROS leading to the disruption of the testicular junction proteins, the activation of germ cell apoptosis and impairment of spermatogenesis [47]. These explanations are in agreement with our resulted histopathological, which showed reduced seminiferous tubular diameters and increased space between the seminiferous tubules in CP treated group. Nrf2 was markedly downregulated by CP resulting in a significant reduction in the activity of GSH and a significantly increased in MDA level subsequently suppress the antioxidant testicular defense activity and these results are consistent with those mentioned by [48]. Administration of NCF to CP-intoxicated rats led to a significant increase in the Nrf2 level accompanied by increased activity of GSH and a reduction in MDA levels, resulting in the increase of antioxidant state in testicular tissues compared to the CP treatment alone. These observations are in accordance with the developing indication for the antioxidant actions of TT which was found to increase Nrf2 and HO-1 and decrease Nf-ƙB and MDA levels in the testicular tissues [49]. It was reported that β-carotene of SP may scavenge free radicals generated by HgCl_2_, thus reducing lipid peroxidation through quenching of singlet oxygen, free radical scavenging and prevent chain breakage during lipid peroxidation [50]. It was reported that omega-3 FA (EPA and DHA) have a powerful protective effect on oxidative stress by modulating cytosolic Ca+ release, and antioxidant activity through reduction of MDA and increase in superoxide dismutase (SOD) and GSH levels thus reduce ROS and inhibit the toxic effect of doxorubicin in testicular tissues of male rats. Also, pretreatment with FAs largely counteracted the histopathologic damage in testicular tissues induced by doxorubicin and preserved the integrity of spermatogenic structures suggesting that fish n-3 FA could effectively protect the testicular cells from doxorubicin apoptotic injury [11]. The current study also highlighted another possible mechanism by which NCF mitigated the gonadotoxicity induced by CP. [51]. CP-induced testicular toxicity was initiated through the production of ROS, leading to activation of NF-κB-mediated pro-inflammatory mediators like tumor necrosis factor-α (TNF-α), IL-6, cyclooxygenase-2 (COX-2), interleukin-1 (IL-1), intracellular adhesion molecule (ICAM), and inducible nitric oxide synthase (iNOS) which were stimulated by Nrf2 deficiency [52]. Nrf2 plays an important cytoprotective role as an antioxidant and anti-inflammatory effect through inhibition of sertoli cell Nf-Kb proteins exhibiting proapoptotic activity beside its sensitvity to oxidants, antioxidants and intracellular redox conditions [53]. On the contrary, the level of NF-κβ and caspase 3 decreased in CP treated with NCF compared to the CP group by about 0.27- and 0.42-fold respectively. Germ cell apoptosis exhibit an essential role in CP-induced testicular damages. Apoptosis is reported to be induced in irreversible manner via activation of caspase 3. For this reason, the apoptotic cells were evaluated using caspase 3 expression [40]. Caspase-3 activity can be served as an essential marker for sperm survival and fertilization capability. Higher caspase-3 activity and DNA fragmentation indicate are detected in sperm with low motility and exposure of phosphatidylserine. So, apoptosis markers including caspase-3 can be used as diagnostic indicators for sperm fertilizability [54]. In this experiment, we found that NF-κB and caspase-3 levels increased, but Nrf2 level decreased in the CP treatment group. Interestingly, NCF markedly mitigated testicular apoptotic event through the reduction of caspase3. Many studies which in accordance with our result showed that TT prevented apoptosis in seminiferous tubules through affecting mRNA expression levels of P53, caspase-3 as well as the Bax/Bcl-2 ratio and thus protects testicular germ cells in CP induced testicular apoptosis in male rats [55]. SP extract was reported to block cytokine treatment induced apoptosis through suppression of ROS production to attenuate oxidative stress and inhibition of apoptosis via a caspase-3-mediated pathway in RINm5F cells of pancreatic β-cells and this activity may be attributed to the presence of high amounts of proteins, all essential amino acids, vitamins, and the strong antioxidant contents of c-phycocyanin and allophycocyanin essential FAs, minerals [56]. SP was reported to reduce the elevated levels of hepatic caspase-3 and TNF-α detected in the lead acetate induced hepatic injury in rats and this effect might be attributed to the elimination of ROS [57]. It was the first time to investigate the effect of SP and FO on the apoptotic status (caspase-3) on CP induce testicular toxicity in male rats which cause a significant reduction from CP group for about 0.65-fold. These results can explain the observed elimination of the disturbed spermatogenesis. From the above-mentioned outcomes, it could be assumed that the combined effect of the three tested ingredients in the optimized NCF enhanced the protective and the treatment efficiency against CP induced testicular toxicity over each individual ingredient. Numerous parameters might contribute to the boosted efficacy of the optimized formulation (SNESNS) but a crucial factor which could elucidate this improvement can be attributed to the presence of three herbal combination of different classes of chemical compounds and thus different mechanism of actions alongside the achievement of a nano-sized formulation to be in the nano range served in enhancing the SP solubility and hence the release and the bioavailability of the drug [58].

## 3. Material and Methods

### 3.1. Plant Material

SP powder was purchased from Imtenan Stores, Egypt. TT dried seed extract was purchased from Nerhado for Industries, Egypt. FO was purchased from Safe Group for Pharmaceutical industries, Egypt.

### 3.2. Chemicals

HPLC grade acetonitrile, dichloromethane, methanol, cyclohexane, n-hexane and castor oil, ethoxylated, pH range 6–8, Acros, organics were purchased from Fisher Chemical (Loughborough, UK). Trifluoroacetic acid (TFA), sulfuric acid, and ortho-phosphoric acid were obtained from Sigma-Aldrich (Chemie GmbH, Germany). All other chemicals were of analytical grade unless otherwise noted. Folin Ciocalteu in the assay of total polyphenol was purchased from Merck KGaA (Darmstadt, Germany). Then, 2, 2-Diphenyl-1-picrylhydrazyl (DPPH), PEG 400, Tween 20 (HLB = 16.7), Tween 80 (HLB = 15), Span 80 (HLB = 4.3), ethanol, propylene glycol, and isopropyl alcohol were obtained from Sigma-Aldrich GmbH (Darmstadt, Germany). Labrasol and Transcutol were kindly obtained as a gift from Gattefosse (St Priest, France). The TLC plate was purchased from Merck, Darmstadt, Germany (HPTLC plate precoated with silica gel 60 F 254 (20 × 20 cm).

### 3.3. Drugs

Tribe Gold^®^ capsule (each capsule contains 250 mg TT) was purchased from local pharmacies and used as a reference drug for evaluation of NCF. Unistin^®^ vials (10 mg CP/10 mL), purchased from Hikma Specialized Pharmaceuticals, Egypt.

### 3.4. Reference Standards

FA was purchased from Sigma-Aldrich Chemical Co., (St. Louis, MO, USA). β-carotene was purchased from Sedico Pharmaceutical company, Egypt. Gallic acid and rutin were purchased from MEPACO Pharmaceutical company, Al-Sharquia, Egypt. TT working standard extract (45% steroidal saponins) was purchased from Nerhado Pharmaceutical company, Giza, Egypt.

### 3.5. Animals

Adult male Wistar rats (200–250 gm) were obtained from the Animal House Colony of the National Organization for Drug Control and Research (NODCAR; Cairo, Egypt). They were housed in polypropylene cages (5 rats/cage) with wire covers and hardwood chips bedding, under specific pathogen-free conditions in facilities maintained at controlled environmental conditions (21–24 °C and 40–60% humidity), and equal light-dark cycles (12/12 h light/dark cycle). The bedding was changed daily in the morning after moving the rats to a new clean cage to avoid the offensive odor. Rats had free access to standard rodent chow and water ad libitum for one week before the onset of the experiment.

### 3.6. Ethics Statement

Experimental design and animal handling were carried out according to the Guide for the Care and Use of Laboratory Animals (NIH, 1996) and after the approval of the Ethics Committee of Faculty of Pharmacy, Cairo University (Cairo, Egypt; PG:3.7.1., MP (2069).

### 3.7. Phytochemical Investigation of SP

#### 3.7.1. Determination of Lipid Content by GC/MS

The preparation of the hydrocarbons was performed according to [59], and methylation of free FAs according to [60]. The chromatographic analysis was performed on Agilent gas chromatography (7890B) coupled with a mass spectrometer detector (5977A). The GC was equipped with HP-5MS column (30 m × 0.25 mm internal diameter and 0.25 μm film thickness), injection volume: 1µL, carrier gas: helium, flow rate: 1ml/min [61].

#### 3.7.2. Investigation of Amino Acid Profile

The acid hydrolyzed amino acids by amide bond breakage were determined according to [62]. The amino acids in SP could be calculated from the following equation [Mg/g dry weight = ppm × dil/wt × 1000]; where, Ppm = concentration of amino acid prepared (mg/L), Dil. = final dilution of sample (mg/mL), Wt. = weight of the sample (mg) using LC 3000 Eppendorf with hydrolysate column SYKAM (S4300).

#### 3.7.3. Total Phenolic Content (TPC) and Total Flavonoid Content (TFC)

TPC was determined using the Folin Ciocalteu colorimetric method [63] with slight modifications. A calibration curve was prepared using a standard solution of gallic acid (0.04, 0.12, 0.2, and 0.28 mg/mL with R^2^ = 0.9988), the samples were performed in triplicates. The standard calibration curve of gallic acid is a linear straight line expressed by the equation y = 3.1973x, R2 = 0.9988 and a slope equals 3.1973 shown in Appendix A. Results were expressed as mg gallic acid equivalent (GAE)/g of the dry powder, using UV-visible spectrophotometer (SPECORD 210 plus), Analytik Jena AG, Germany. TFC was evaluated according to the method described by [64]. A calibration curve was prepared using a standard solution of rutin (0.04, 0.08, 0.12, and 0.16 mg/mL with R^2^ = 0.9995), the samples were performed in triplicate. The standard calibration curve of rutin is a linear straight line expressed by the equation y = 0.0062x, R2 = 0.9992 and a slope equals 0.0062 shown in Appendix A. Results were expressed on a dry weight basis as mg rutin equivalent/g sample.

#### 3.7.4. Determination of Vitamin B12, Vitamin C, and Vitamin E by HPLC

The quantitative estimation of vitamin B12, C, and E was performed for SP according to [65] using HPLC Agilent 1260: with column Kromasil 100-5-C18.

#### 3.7.5. Determination of Mineral Content

Briefly, 0.5 g of SP was wet digested using a sulphuric-perchloric-acids mixture according to the procedure of [66]. The total content of iron, zinc, and copper was determined by inductively coupled plasma spectrometry (ICP) (Ultima 2 JY Plasma).

### 3.8. Formulation of a Novel Herbal SNESNS

#### 3.8.1. Preparation of SNEDDS

##### Screening of Surfactants and Co-Surfactants for SNEDDS

Several surfactants namely; Labrasol, Cremophor EL, Tween 80, Tween 20, Transcutol, and Span 80 and four types of cosurfactants namely; ethanol, isopropyl alcohol, PEG 400, propylene glycol were investigated to select the most suitable surfactant and co-surfactant with FO. 10 μL of FO was added to 1 g of each surfactant or co-surfactant separately with vigorous vortexing for 2 min. If a clear solution was attained, the oil addition was repeated until a hazy solution was reached [67]. A mixture of the best two surfactants was tested in different concentrations, namely 1:1, 2:1, 3:1, 1:2, and 1:3. The tested surfactant and co-surfactant that can incorporate more oil were selected for further investigation.

##### Construction of Ternary Phase Diagram

The ternary phase diagram was constructed using Sigmaplot^®^ software (Systat Software Inc., CA, USA, http://www.sigmaplot.co.uk/products/sigmaplot/sigmaplot-details.php); each triangle head corresponds to 100% of each component. Different weight ratios of FO, the selected surfactant, and cosurfactant were used to define the margins of phases formed as presented in Appendix A. The ternary mixtures were shaken at 200 rpm at 40 °C utilizing an isothermal shaker till complete mixing. Afterward, 1 g of each system was stirred in 250 mL distilled water at 37 °C where each system was inspected visually [68]. A SNEDDS can be accepted if it is rapidly emulsified in the aqueous phase giving a transparent or bluish nanoemulsion [69].

#### 3.8.2. Characterization of SNEDDS

##### Assessment of Emulsification Time (ET), Dispersibility, and %Transmittance (%T)

ET is considered a helpful method to postulate the emulsification efficiency of the SNEDDS. The test was performed visually using USP dissolution type II apparatus (Hanson Research, Chatsworth, CA, USA) [70]. One gram of the tested system was placed into distilled water (500 mL) using 50 rpm at 37 °C. ET was noted as the time taken to achieve a transparent solution. The following grading was denoted to conclude the efficiency of self-emulsification development [71,72]:Grade A: a transparent or bluish nanoemulsion formed within 1 min of mixing.Grade B: translucent or bluish-white nanoemulsion formed within 1 min of mixing.Grade C: milky emulsion formed within 2 min.Grade D: greyish white or slightly oily emulsion developed in more than 2 min.Grade E: a system that does not display satisfactory self-emulsification resulting in large visible oil droplets.

Systems showing superior emulsification efficiency (Grade A) were selected for additional exploration.

%T can be considered a predictive tool for the optical transparency of the nanoemulsions. % T of the developed systems was measured using UV Shimadzu spectrophotometer (Kyoto, Japan) at 650 nm with distilled water as a blank [68].

##### Simulation of the Physiological Dilution Process after Oral Administration

Dilution robustness was accomplished to mimic the physiological dilution progression after the oral administration of the developed systems. Visual evaluation of dilution (50, 100, and 1000-fold) of the formulated SNEDDSs with several media namely; distilled water, 0.1 N HCl, and phosphate buffer pH 6.4 was performed for the detection of any physical instability [68,71].

##### Globule Size(GS) Analysis, Polydispersity Index (PDI), and Zeta Potential (ZP)

The GS and ZP are considered key parameters for the stability of the nanoemulsion. Decreasing the droplet size can promote both the absorption and the bioavailability of the investigated herbal extracts [73]. The average GS together with the PDI and ZP of the nanoemulsions were measured by Malvern zeta sizer (Nano ZS90, Malvern Instrument Ltd., Worcestershire, UK). The selected sample was 50-fold diluted with double distilled water and the testing was done in triplicates [14,74].

#### 3.8.3. Preparation of the Extract Loaded SNESNS

Extract loaded SNESNS was developed to optimize the amount of SP content in the SNEDDS pre-concentrate by homogenizing the required weight of SP into the selected SNEDDS at 19,000 rpm for 30 min at 25 °C to achieve uniform dispersion. The SP was dispersed into the selected SNEDDS as both solubilized molecules and suspended microparticles. Afterward, SP SNEDDS was stirred into TT aqueous solution to form a nanoemulsion which spontaneously develops a nanosuspension (nanoemulsion-based nanosuspension, SNESNS).

#### 3.8.4. Characterization of the Extract Loaded SNESNS

##### Particle Size (PS), Polydispersity Index (PDI), and Zeta Potential (ZP)

The average PS together with the PDI and ZP of the SNESNS were measured using the Malvern zeta sizer (Nano ZS90, Malvern Instrument Ltd., Worcestershire, UK). SNESNS was 200-fold diluted with double distilled water and the testing was performed in triplicates [75].

##### Transmission Electron Microscopy (TEM)

The morphology of a nanosuspension is a vital parameter to be studied. The morphology of the extract-loaded SNESNS diluted in the ratio of 1:200 (*w*/*v*) was investigated. A drop of the tested sample was positioned on a grid stained with 2% phosphotungstic acid solution with the removal of excess stain. Finally, the grid was left to dry at room temperature before taking the TEM images [28,76].

### 3.9. Quantitative Analysis and Method Validation of NCF

#### 3.9.1. Quantitative Analysis of GC/FID Assay of FAME in NCF and its Ingredients

The analysis of FAME in SP, FO, TT, and NCF was proceeded according to [77] using gas chromatography coupled with flame ionization detector (FID), as well as 30 m long, 0.32 mm. diameter, and 0.25 µm film thickness, cross-linked 5% phenyl polysiloxane (HP5- capillary column, Hewlett Packard, Palo Alto, CA, USA) fused-silica column.

##### Standard Preparation

Standard calibration curve for FAs standards were prepared at expected concentrations of 0.72, 1.44, 2.16 and 2.88 mg/mL palmitic acid, 0.136, 0.272, 0.408 and 0.544 mg/mL GLA, 0.8, 1.6, 2.4 and 3.2 mg/mL LA, 0.0956, 0.1912, 0.2868 and 0.3824 mg/mL EPA and 0.1356, 0.2712, 0.4068 and 0.5424 mg/mL DHA in n-hexane.

##### Sample Preparation

The raw materials and NFC were prepared at concentrations of 20 mg/mL SP, 2 mg/mL TT, 8 mg/mL FO, and 88 mg/mL NCF and subjected to methylation.

#### 3.9.2. Quantitative Analysis of β-Carotene in SP and NCF by HPLC and HPTLC

HPLC investigation: The method was proceeded according to the method that was validated for assay of β-carotene in wheat germ oil sent from Sedico Pharmaceutical Industry using HPLC apparatus Thermo (Dionex), with column ODS Hypersil C18 (Length 150 mm × ID 4.6 mm; particle size: 5 µm). HPTLC investigation: 10 µL from both samples and the standard solutions were injected using precoated silica gel plates as stationary phase, and allowed to develop for a distance of 10 cm using the mobile phase system (S3) consisting of (petroleum ether: cyclohexane: ethyl acetate: acetone: ethanol) (60: 16: 10: 10: 6) [78] and measured at 366 nm by TLC densitometer using HPTLC apparatus (CAMAG, Muttenz, Switzerland): which consists of Camag microliter syringe, sample applicator (Linomat V), CAMAG twin trough glass tank, densitometric Camag TLC scanner VI operated by Win CATS software and the source of radiation utilized were deuterium and tungsten lamp.

##### Standard Preparation

Standard calibration curve was prepared at expected concentrations of 0.004, 0.008, 0.012, and 0.016 mg/mL for HPLC investigation and 0.04, 0.08, 0.12, and 0.16 mg/mL β-carotene in cyclohexane for HPTLC investigation in cyclohexane.

##### Sample Preparation

Both the SP and NCF were prepared at concentrations of 40 mg/mL for SP and 47.2 mg/mL (equal to 11.88 mg SP/mL) of NCF in cyclohexane using light protected volumetric flasks.

#### 3.9.3. UV Colorimetric Investigation of Total Steroidal Saponins in TT and NCF

The method was proceeded according to [79] which is based on measuring the intensity of the color developed when the steroidal saponins react with *p*- dimethylaminobenzaldehyde (p-DMAB) in presence of concentrated hydrochloric acid at high temperature.

##### Standard Preparation

The standard calibration curve of TT working standard extract was prepared at expected concentrations of 2, 4, 6, and 8 mg/mL steroidal saponins of TT extract in methanol.

##### Sample Preparation

Both the TT extract and SNESNS were prepared at a concentration of 3 mg/mL and 1.041 mg/mL in methanol, From the standard and the samples, 3 mL will be taken to 10 mL volumetric flask, add 5 mL of modified Ehrlich reagent (1 gm p-DMAB added to 34 mL concentrated hydrochloric acid and complete the volume to 100 mL with methanol), then the mixture was refluxed in a water bath for 2 h, allowed to cool, and completed the volume to 10 mL with methanol. Perform the blank by mixing equal volumes of methanol and reagent. For NCF, perform the placebo by dissolving 347.6 mg SP and 139.04 mg FO in 10 mL methanol then perform defatting of the chlorophyll of SP using diethyl ether to exclude the color of them and measure at 515 nm.

## 4. Biological Evaluation of the NCF

### 4.1. Experimental Design

Sixty rats were allocated randomly into six groups. The first group represented the normal control group (NC) that was administrated saline orally. Testicular toxicity was induced by a single intraperitoneal (IP) dose of CP (7 mg/kg) [80] on day 11th. Rats were treated for 20 consecutive days, 10 days before, and 10 days after the administration of CP. The second group included the untreated rats that were administrated CP only (NC group). In groups 3, 4, 5, and 6, rats treated orally with TT at a dose of (100 mg/kg/day) [81]; SP at a dose of (1000 mg/kg/day) [82]; FO at a dose of (400 mg/kg/day) and NCF group at a dose of (4.4 g/kg/day). The last dose of any treatment was given 24 h before the end of the experiment.

### 4.2. Sperm Count and Motility

Testes and epididymis were immediately removed and cleaned from the adhering tissue and weighed. The epididymal content of each rat was obtained after cutting the tail of the epididymis and squeezing it gently and examined for sperm count, presence of abnormalities, motility, and life-dead percentage according to the technique adopted from [83].

### 4.3. Histopathology

Tissue specimens from testes seminal vesicles of both control and treated animals were fixed in 10% neutral buffered formalin for 24 h, dehydrated in gradual ethanol (70–100%), cleared in xylene, and embedded in molten paraffin at 58–60 ℃. Five-micron thick paraffin sections were prepared and then routinely stained with hematoxylin and eosin (H and E) dyes according to [84].

### 4.4. In Vitro Antioxidant Activity (DPPH Radical Scavenging Activity)

The in vitro antioxidant activity for SP and NCF was performed according to [85]. The DPPH was prepared at a concentration of 0.01% in ethanol (HPLC grade). The SP was dissolved in absolute ethanol and at different concentrations of 0.18, 0.36, 0.54, 0.72, and 0.9 mg/mL, and equivalent concentrations from NCF were prepared as mentioned above then 1 mL of each test solution was added to 6 mL of DPPH solution. A blank sample was run using absolute ethanol. The test solutions were incubated for 30 min. at room temperature and the absorbance was recorded against the blank at 517 nm. Gallic acid was used as a positive control at a concentration of 0.01, 0.02, 0.03, and 0.04 mg/mL. The ability to scavenge the DPPH radical was expressed as percentage inhibition and calculated using the following equation: DPPH scavenging activity (%) = (A_blank_ − A_sample_)/A_blank_] × 100; where A_blank_: the absorbance of the blank and A_sample_: the absorbance of the sample.

### 4.5. Biochemical Analysis

At the end of the experimental period, rats were anesthetized using thiopental (50 mg/kg, IP). Blood samples were collected from retro-orbital sinus then centrifuged to separate serum and kept at −80 °C till the determination of testosterone level using Rat Testosterone ELISA kit (Cusabio, PRC). Animals were weighed and sacrificed by cervical dislocation under anesthesia. Dead animals were frozen till their incineration. The right testis was decapsulated and homogenized in ice-cold 0.05 mM phosphate buffer (pH 7.4) to prepare 10% homogenate and the resultant supernatant were stored at −80 °C till estimations of determination of biochemical parameters. Testicular oxidative stress and antioxidant markers, malondialdehyde (MDA), as an index for lipid peroxidation; reduced glutathione (GSH); total antioxidant capacity (TAC) content; were measured using the Biodiagnostic Colorimetric Kits (Cairo, Egypt) according to the manufacturer’s instruction. In another aliquot, ELISA rat kits were used for the estimation of testicular Nuclear Factor [Erythroid-Derived 2]-Like 2 (Nrf2) (MyBioSource, Inc., San Diego, CA, USA; Cat #: MBS752046). Nuclear Factor-Kappa B (NF-κB) (MyBioSource, Inc., San Diego, CA, USA; Cat #: MBS268833), and Interleukin 6 (IL-6) (Wuhan Fine Biotech Co., Ltd., Wuhan, Hubei, China; Cat #: ER0042) were assessed to determine testicular inflammatory status. Caspase 3 was estimated to evaluate the apoptotic event (MyBioSource, Inc., San Diego, CA, USA; Cat #: MBS743552). The above parameters were assayed according to the manufacturer’s instructions of the related ELISA kit.

### 4.6. Statistical Analysis

GraphPadPrismv5.0 (GraphPad Prism Inc., La Jolla, CA, USA) was used to analyze and present all the data. Data were expressed as mean ± SD of six animals. Statistical comparisons between means were carried out using one-way analysis of variance (ANOVA), followed by the Student–Newman–Keuls test. The statistical significance of difference was considered at *p* < 0.05.

## 5. Conclusions

In conclusion, the current study provides evidence for the promising ameliorative effects of SNESNS on gonadotoxicity induced by CP. In the current study, the design of an NCF with SNESNS has been applied to mitigate or minimize CP-induced spermatotoxicy. This is the first design of a unique formulation of three variable nutraceutical origins with different physical character in a homogenous stable SNESNS drug delivery system, and each 1 g of the prepared formula is composed of 250 mg SP which is blue-green algae, 100 mg FO which is of fish source, and TT as a dried herb extract using Span 80/cremophor EL as surfactant and isopropanol as cosurfactant in the establishment of SNEDDS with FO in the percentage of (30% FO; 50% span80/cremophor EL; 20% isopropanol). Further, the SP is loaded in the previous system alongside with TT aqueous solution to form a dark green, homogenous, stable semisolid mixture with potential solubility and thus bioavailability. In the current study, The NCF of SNESNS design is used for the first time as an adjuvant supplement to diminish the toxic side effect of CP, especially in testicular tissues. Also, it is the first time to incorporate SP in pharmaceutical dosage form and a novel design (SNESNS) for treatment of spermatotoxicity. Analytical methods development and validation were employed to control the specific properties of the NCF. The same approach was applied for each ingredient comprised in the NCF. Strikingly, NCF showed the best near-to-normal histological and spermatogenic features comparing to the control group, suggesting the efficacy of NCF as a new therapeutic approach for the protection of the male reproductive system from the destructive toxic effects of CP. Additionally, TT, FO, and SP groups providing promising ameliorative effects of NCF on gonadotoxicity induced by CP. These beneficial actions were mediated, at least partly, via intervention with antioxidative, anti-inflammatory, and antiapoptotic events via disruption of Nrf2/NF-κB/caspase-3 cross-talk. The combination of the nutraceutical therapy with the newly designed formula of the SNESNS drug delivery system showed a synergistic effect with enhanced efficacy using poorly soluble ingredients. Further studies are necessary to clarify the exact molecular mechanisms for the NCF. Moreover, future clinical studies are recommended to investigate the role of this new formulation as a dietary supplement involved in the chemotherapeutic protocol including CP as chemotherapy for young men with carcinoma.

## Figures and Tables

**Figure 1 molecules-26-00408-f001:**
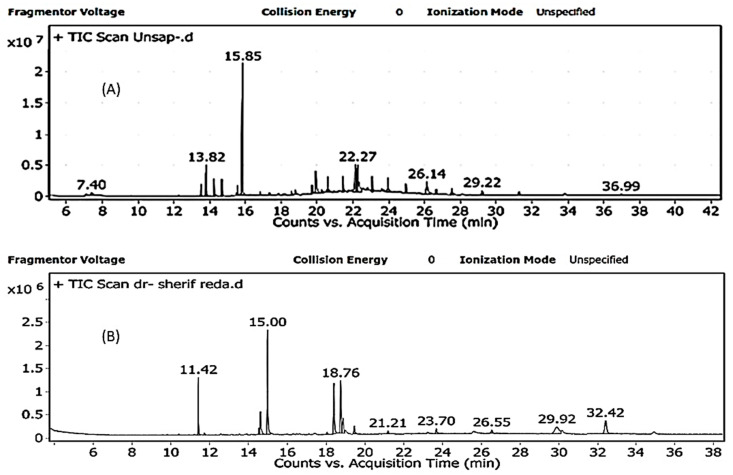
GC/MS of (**A**) USM with heptadecane as the main hydrocarbon and (**B**) fatty acid methyl esters (FAME) with palmitic acid as the main FA of SP.

**Figure 2 molecules-26-00408-f002:**
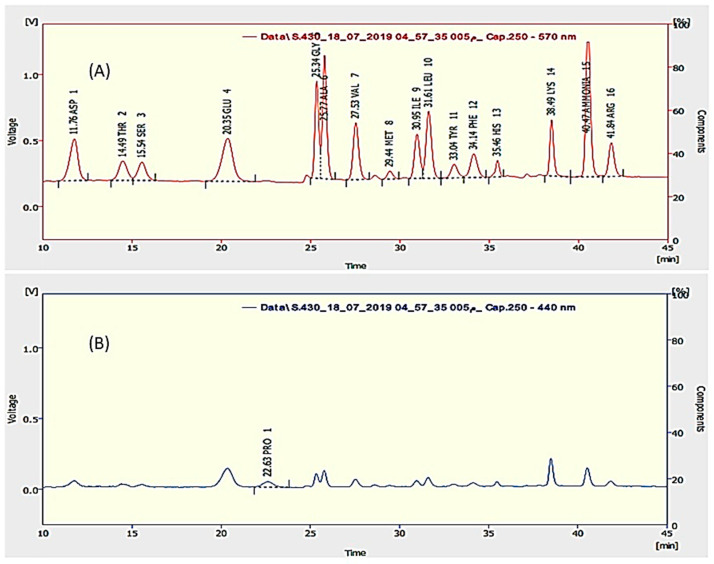
HPLC chromatogram of amino acids in SP at λ 570 nm (**A**) and for proline at λ 440 nm (**B**)**.**

**Figure 3 molecules-26-00408-f003:**
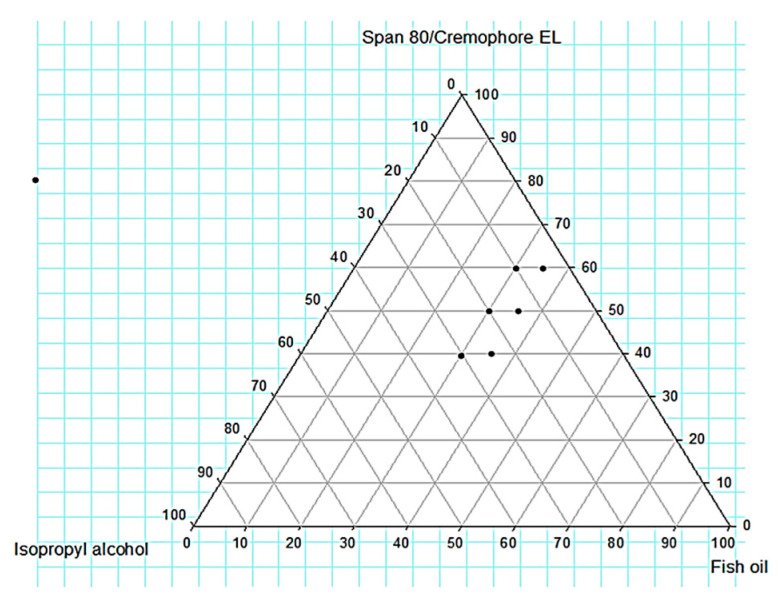
Pseudo-ternary phase diagrams of SNEDD of different compositions of the selected System (FO, span 80/cremophor EL and isopropyl alcohol).

**Figure 4 molecules-26-00408-f004:**
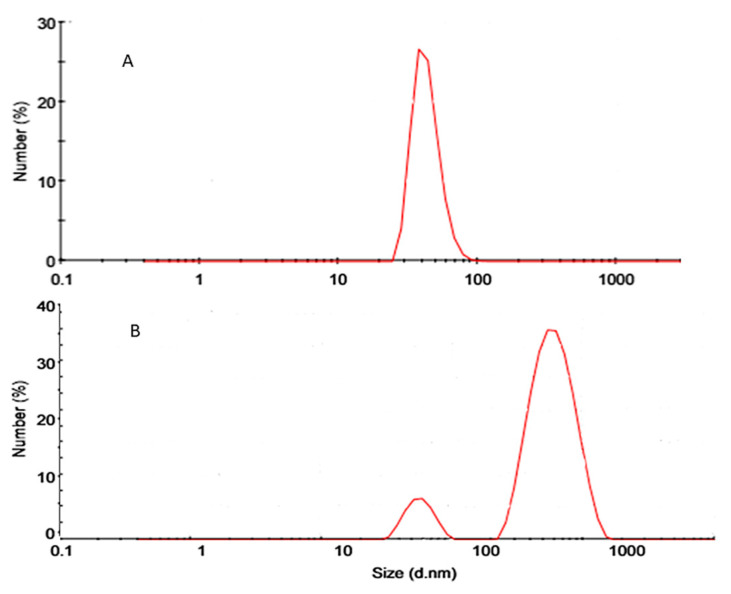
Particle size distribution (**A**) for the selected SNEDDS (**B**) for the extract-loaded SNESNS.

**Figure 5 molecules-26-00408-f005:**
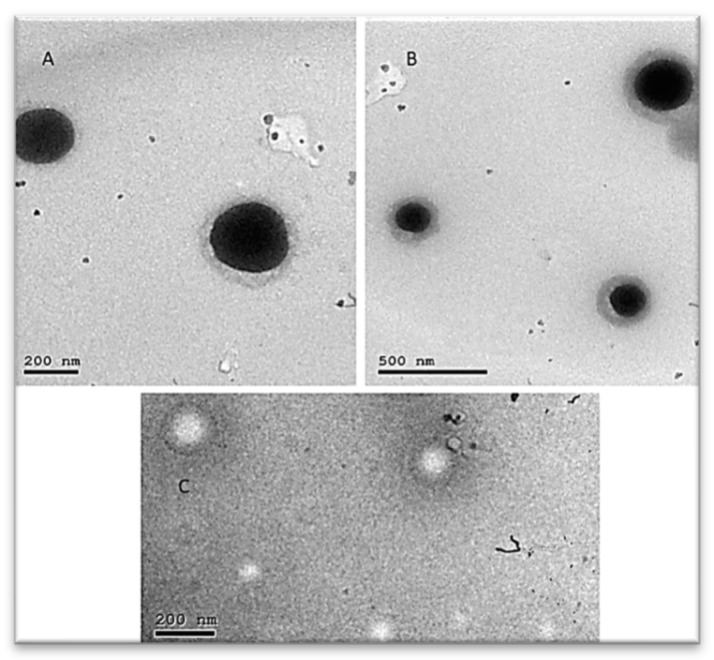
TEM of D-SNESNS (**A**,**B**): NENS formed by diluting D-SNESNS with distilled water showing spherical (**C**) nanoemulsion globules formed by dilution of SNEDDS with water.

**Figure 6 molecules-26-00408-f006:**
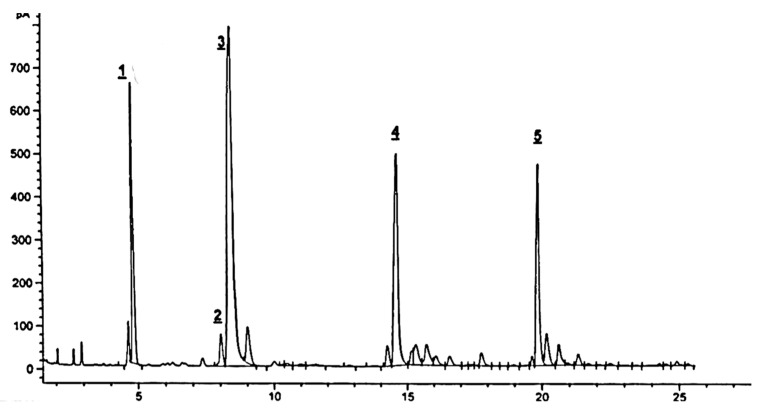
GC/FID chromatogram of FAME in NCF; 1: palmitic acid; 2: GLA; 3: LA; 4: EPA; 5: DHA.

**Figure 7 molecules-26-00408-f007:**
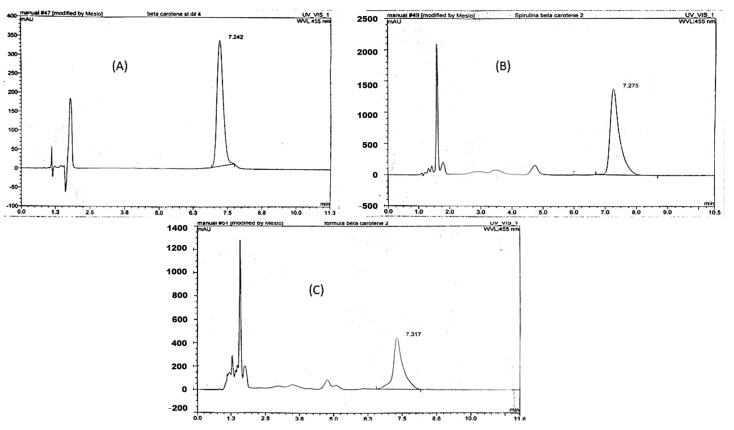
HPLC chromatogram of β-carotene in (**A**): reference standard; (**B**): SP and (**C**): NCF.

**Figure 8 molecules-26-00408-f008:**
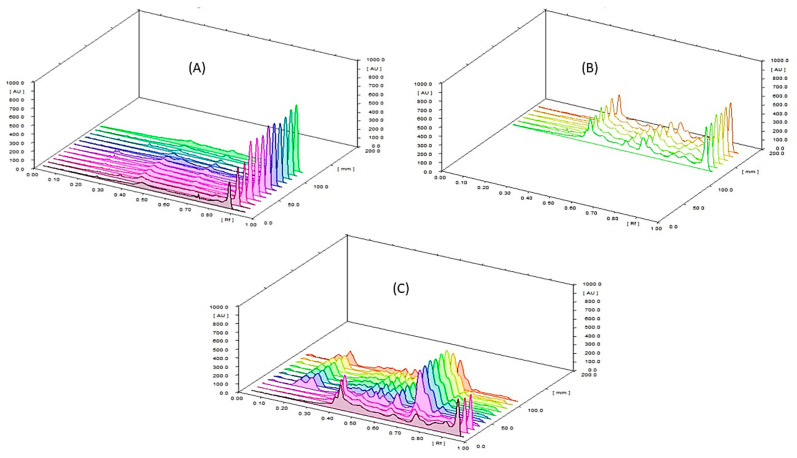
HPTLC chromatogram of β-carotene in (**A**): reference standard; (**B**): SP and (**C**): NCF (the first three pink tracks represent the β-carotene for SP while the rest of the tracks represent NCF).

**Figure 9 molecules-26-00408-f009:**
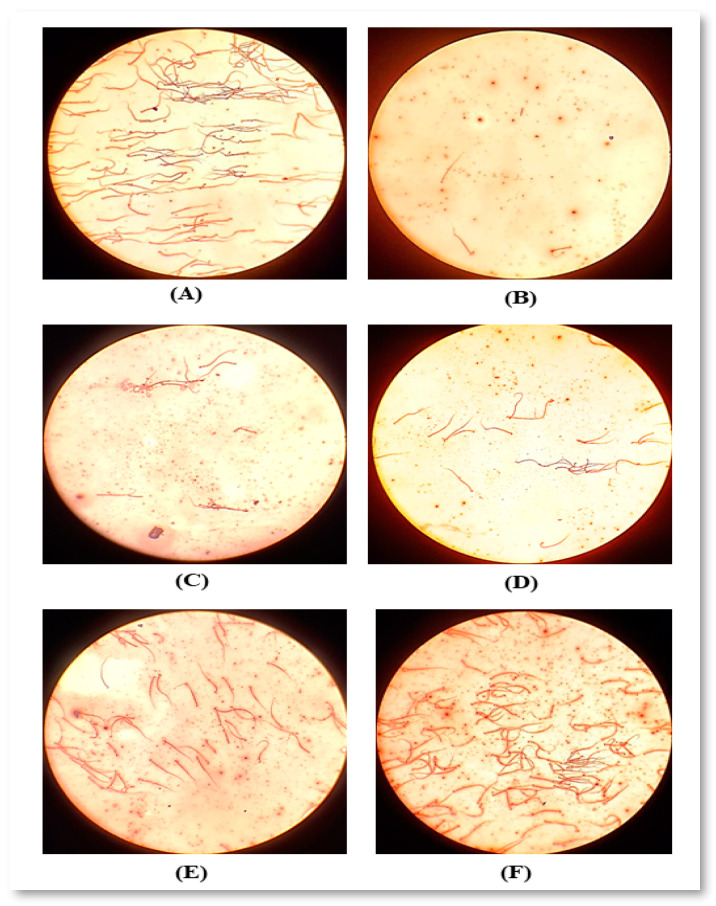
Semen investigation under a light microscope for (**A**): NC, (**B**): CP, (**C**): FO, (**D**): TT; (**E**): SP; (**F**): NCF.

**Figure 10 molecules-26-00408-f010:**
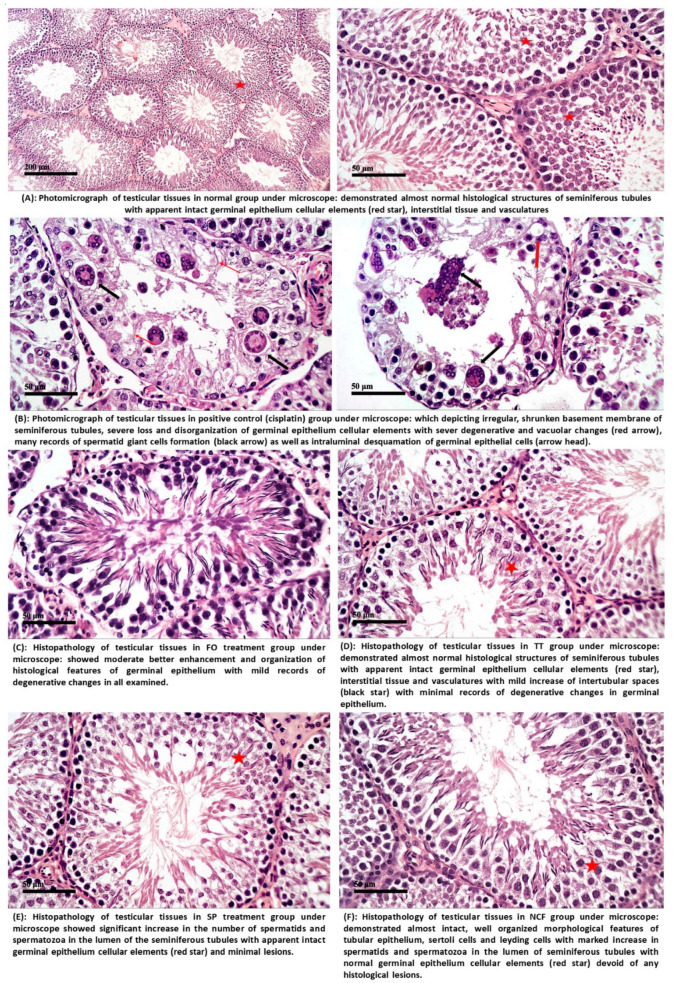
Histopathological estimation for; (**A**): NC, (**B**): CP, (**C**): FO, (**D**): TT; (**E**): SP; (**F**): NCF.

**Figure 11 molecules-26-00408-f011:**
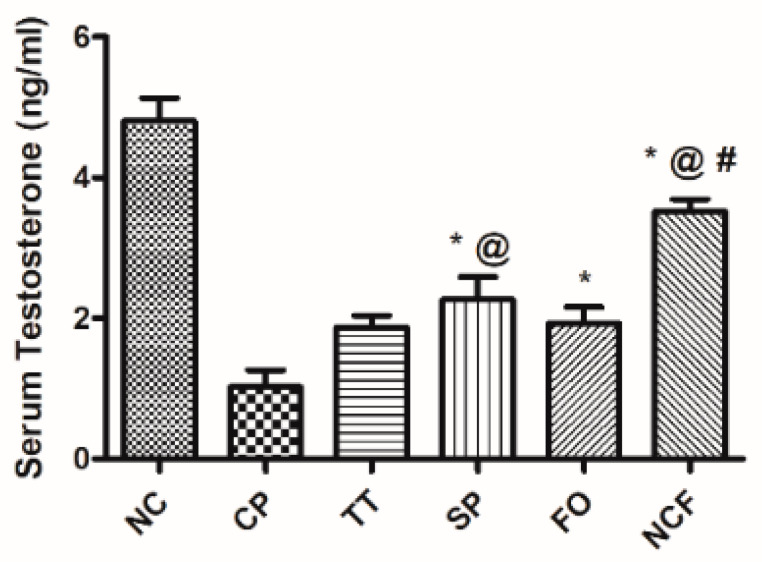
Effect of TT, SP, FO, and NCF on cisplatin-induced alterations in serum testosterone in adult male rats. Effect of 20 days administration of TT (100 mg/kg/orally); SP (1000 mg/kg/orally); FO (400 mg/kg/orally); and new nutraceutical formula (NCF) at a dose of (4.4 g/kg/orally) on serum testosterone. Values are means of 6–10 rats ± SD. Statistical analysis was performed using one-way ANOVA followed by Newman Keuls multiple comparison tests (*p* < 0.05). As compared with normal control (NC; *), cisplatin (CP; @), *Tribulus terrestris* (TT; #). *Spirulina platensis* (SP), Fish oil (FO).

**Figure 12 molecules-26-00408-f012:**
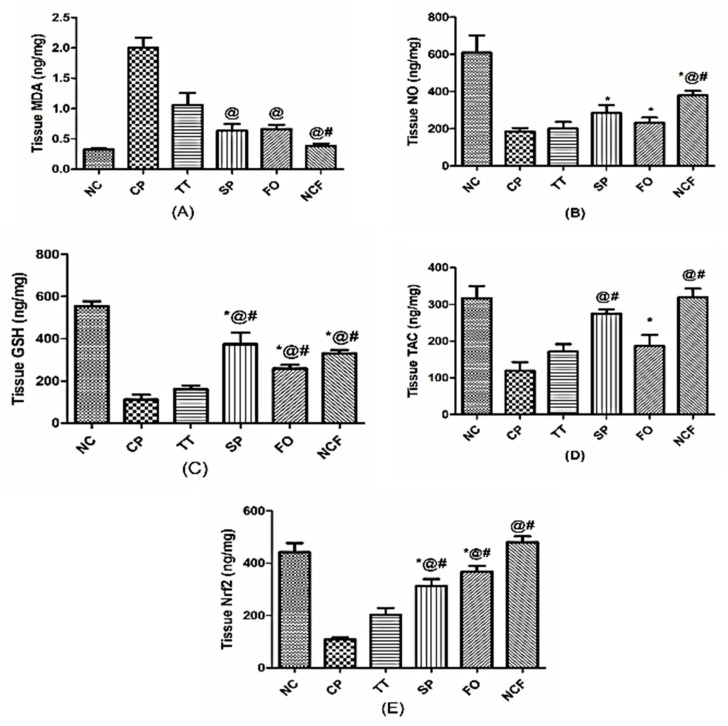
Effect of TT, SP, FO, and NCF on cisplatin-induced alterations in testicular oxidative stress and antioxidant markers in adult male rats. Effect of 20 days administration of TT (100 mg/kg/orally); SP (1000 mg/kg/orally); FO (400 mg/kg/orally); and new nutraceutical formula (NCF) at a dose of (4.4 g/kg/orally) on testicular content of (**A**) MDA, (**B**) NO, (**C**) GSH, (**D**) TAC, and (E) Nrf2 in CP-untreated rats. Values are means of 6-10 rats ± SD. Statistical analysis was performed using one-way ANOVA followed by Newman Keuls multiple comparison tests (*p* < 0.05). As compared with normal control (NC; *), cisplatin (CP; @), *Tribulus terrestris* (TT; #). *Spirulina platensis* (SP), Fish oil (FO), malondialdehyde (MDA), reduced glutathione (GSH); total antioxidant capacity (TAC), Nuclear Factor [Erythroid-Derived 2]-Like 2 (Nrf2).

**Figure 13 molecules-26-00408-f013:**
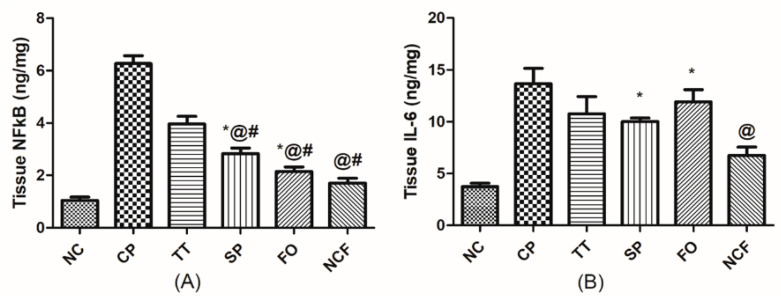
Effect of TT, SP, FO, and NCF on cisplatin-induced alterations in testicular inflammatory status in adult male rats. Effect of 20 days administration of TT (100 mg/kg/orally); SP (1000 mg/kg/orally); FO (400 mg/kg/orally); and new nutraceutical formula (NCF) at a dose of (4.4 g/kg/orally) on testicular content of (**A**) NF-κB and (**B**) IL-6 in CP-untreated rats. Values are means of 6-10 rats ± SD. Statistical analysis was performed using one-way ANOVA followed by Newman Keuls multiple comparison tests (*p*< 0.05). As compared with normal control (NC; *), cisplatin (CP; @), *Tribulus terrestris* (TT; #). *Spirulina platensis* (SP), Fish oil (FO), Nuclear Factor-Kappa B (NF-κB), Interleukin 6 (IL-6).

**Figure 14 molecules-26-00408-f014:**
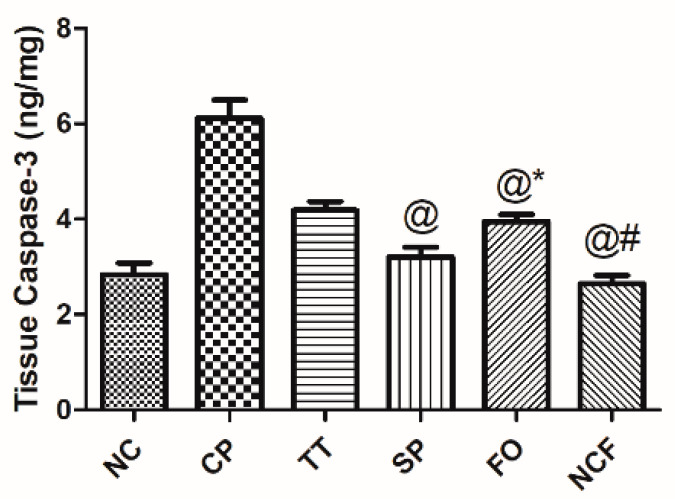
Effect of TT, SP, FO, and NCF on cisplatin-induced alterations in testicular caspase-3 in adult male rats. Effect of 20 days’ administration of TT (100 mg/kg/orally); SP (1000 mg/kg/orally); FO (400 mg/kg/orally); and new nutraceutical formula (NCF) at a dose of (4.4 g/kg/orally) on testicular content of caspase in CP-untreated rats. Values are means of 6-10 rats ± SD. Statistical analysis was performed using one-way ANOVA followed by Newman Keuls multiple comparison tests (*p*< 0.05). As compared with normal control (NC; *), cisplatin (CP; @), *Tribulus terrestris* (TT; #). *Spirulina platensis* (SP), Fish oil (FO).

**Table 1 molecules-26-00408-t001:** The nutritional value of the SP.

Chemical/Element	Content in SP	Chemical/Element	Content in SP
Essential amino acids (mg/g):		Total protein content (%)	68.77
Leucine	63.94	Total flavonoid content (mg RE/g DW)	0.864
Valine	52.56	Total polyphenol content (mg GAE/g DW)	58.64
Isoleucine	45.62	Total antioxidant activity (%)	25.4
Phenyl Alanine	41.71	Vitamins (mg/100 g):	
Threonine	33.87	Cyanocobalamin (Vit. B_12_)	3.2
Lysine	27.71	Ascorbic acid (Vit. C)	191.906
Histidine	12	α-Tocopherol (Vit. E)	undetectable
Methionine	10.14	β-Carotene(Vit. A)	44
Non-essential amino acids (mg/g):		Minerals (mg/100 g):	
Glutamate	141.58	Iron (Fe)	170
Aspartic Acid	85.23	Zinc (Zn)	160
Arginine	48.91	Copper (Cu)	20
Glycine	38.59	Fatty acid methyl esters (FAME) (mg/g):	
Alanine	27.07	Palmitic acid	132
Tyrosine	22.99	Gamma-linolenic acid (GLA)	58.74
Serine	21.58	Linoleic acid	27.7
Proline	14.2	Total fatty acid content (%)	22

**Table 2 molecules-26-00408-t002:** The chemical composition and standardization of the NCF.

Chemical/Element	Content in the New Formula
Each 1 g new formula consisting of:	
*Spirulina* powder	250 mg
*Tribulus terrestris* extract (standardized as 45% steroidal saponin)	50 mg
Omega-3-fish oil (standardized as 50% EPA and DHA)	100 mg
Total antioxidant activity (%)	35.3
β-Carotene (Vit. A) (mg/g)	0.17
Total fatty acid content (%)	27.23
Fatty acid content (mg/g):	
Palmitic acid	110.68
Gamma-linloenic acid (GLA)	16.87
Linoleic acid	104.12
Eicosapentaenoic acid (EPA)	22.04
Docosahexaenoic acid (DHA)	18.58
Content of steroidal saponins of *Tribulus terrestris* extract in new formula (mg steroidal saponin/g formula)	36.3

## Data Availability

All data are availables for cientific community.

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
