# Peer review of "Phytochemical and Biological Evaluation of a Newly Designed Nutraceutical Self-Nanoemulsifying Self-Nanosuspension for Protection and Treatment of Cisplatin Induced Testicular Toxicity in Male Rats"

_molecules, 2021, doi:10.3390/molecules26020408_

Round 1

Reviewer 1 Report

The drug delivery issue and the harmful side effects of chemotherapy drugs are by no means very important challenges faced by the medical and pharmaceutical scientists and practitioners. The authors of the manuscript approach the challenge by investigating self-nanoemulsifying self-nanosuspensions by introducing herbal components which by themselves carry valuable active substances. The results reported might add value to the research carried out in this area. Therefore, the manuscript could be accepted for publication after a serious revision by the authors.

- Abstract is somewhat too long. According to the Instructions for Authors, it should be maximum 200 words.

- My suggestion is to include the information about the composition of the best self-nanoemulsifying self-nanosuspension suggested based on the research described in the manuscript into Abstract.

- The same suggestion is even more important in the Conclusions part. What are the findings of the research carried out? Instead of repeating the same information from Introduction, the emphasis on the results and the best of them should be included into Conclusions. The exact composition of the “newly designed formula of self-nanoemulsifying self-nanosuspension drug delivery system“ should be indicated.

- The necessity of the Abbreviation section should be reconsidered. The two independent systems may be used: either abbreviation is provided in brackets after the full term when it is mentioned for the first time, or the abbreviation list is provided. Of course, with so many abbreviations used in the text, it might be easier for the reader to check the meaning of the abbreviation if needed. However, the expected readers of the article are professionals in the field, thus familiar with the terms.

- Maybe a table representing different SNESNSs with their composition indicated and main characterization parameters provided would add much needed possibility to view the overall extent of the study and main results achieved.

- Author Contributions should be listed as described in the Instructions for Authors.

 - References should be described as requested by the journal –abbreviated journal names instead of full names and no issue number should be provided.

- Authors should check the manuscript critically to unify the terms (self nano-emulsifying self nano-suspension in the title, but self-nanoemulsifying self-nanosuspension or self nanoemulsifying in the manuscript; 50 –fold vs. 50 fold, etc.); regular and bald words „Table, Figure“ in the text. It is necessary to check for missing gaps between words and sentences starting from the lower-case letter. In general, a very careful revision is needed to reach a level of a neat text, which would be suitable for publication.

- The English language of the manuscript requires a very careful revision as well. The manuscript cannot be read with pleasure. In some cases one must force herself/himself to continue reading as the sentence structure is very cumbersome. The authors should revise the tenses, singular/plural nouns and associated verbs. I would suggest writing the information, which has been published already and has already become a well-known truth, in the present perfect tense instead of past tense. Tense in the same paragraph should be checked to match.

Some sentences are very “heavy” and unnatural, for example:

Lines 108-110 – “It was noticed the presence of myristic, heptadecanoic, stearic, oleic, palmitoleic, omega-3, omega-6, linoleic acid  (LA), GLA, and palmitic acid, the highest saturated fatty acid in SP, which was found to be in  accordance with the results of this study [17].“ Maybe it could be replaced with „“Myristic, heptadecanoic, stearic, oleic, palmitoleic, omega-3, omega-6, linoleic acid (LA), GLA, and palmitic acid, the highest saturated fatty acid, were identified in SP. These results are in accordance with the ones in [17].“

Lines 133-134 – „It was the first time to estimate the TFC in commercialized SP in Egypt using rutin as a reference standard.“ Could be replaced with „ For the first time, TFC was estimated in commercial SP in Egypt using rutin as a reference standard.“

Lines 136-140 – The sentences in the paragraphs like this “SP was found to contain (3.2 mg/100 g) cyanocobalamin (vitamin B12) and (191 mg/ 100g) ascorbic acid (vitamin C) while α-tocopherol (vitamin E) was undetectable. It was reported that the SP contain (175 μg vitamin B12 /100 g) and (9.86 mg vitamin E/100 g) [19]. Also, it was found that SP contain (162 138 μg/100 g) [23]. The SP was found to contain (60 mg vitamin E/100 g) [18] and these results were inconsistent with the results of this study as vitamin B12 was more than found in the previous studies and vitamin E was undetectable as shown in Table S3.“ maybe could  be corrected as follows: “SP contained 3.2 mg/100 g of cyanocobalamin (vitamin B12) and 191 mg/ 100g of ascorbic  acid (vitamin C) while α-tocopherol (vitamin E) was undetectable. It was reported that 175 μg/100 g of vitamin B12 and 9.86 mg/100 g of vitamin E were detected in SP [19]. Other studies reported 138 μg/100 g of WHAT [23] and 60 mg/100 g vitamin E in SP [18]. However, these results were inconsistent with the results of this study as the content of vitamin B12 was higher than reported in the previous studies  and vitamin E was undetectable as shown in Table S3.“

Here are some more remarks:

Line 20 – causes.

Line 26 – aside.

Line 41 – CP - Full name must be given when mentioned for the first time.

Lines 44-45 – the sentence should be corrected.

Line 48 and elsewhere– my suggestion would be to replace word “embarrassment” with “retardation” or “obstruction” as the word embarrassment has also the other meaning.

Line 68 - linoleic acid is the main FA in FO.

Lines 155-156 – the sentence should be corrected.

Line 190 – eliminated instead of omitted.

Line 219 – starting with this place abbreviation SP has two meanings, what is not acceptable - suspended drug (SP)

Lines 264-266 – how can the content be calculated by HPLC?

Line 275 – were injected.

Lines 303-305 – the sentence should be corrected to communicate the meaning clearly.

Table 2 – gm should be replaced with g.

Line 348 – Tables.

Line 421-424 – the sentence should be corrected, the tenses should be matched.

Materials and methods section – the word “supplied from” should be replaced with “purchased” or “obtained”. “Supplied by” could be used indicating that some other person or company gave something to the authors and in the line 538.

Line 592 – “Half gram“ should be written as 0.5 g.

Line 602 – was instead of is – tenses should match.

Lines 657-658 – this information is not suitable in the Materials and Methods section, since the exact procedures must be provided here. All discussions should be moved to the Discussion section.

Lines 677 and 680 – my suggestion would be to write just „HPLC investigation“ and „HPTLC investigation“.

Line 679 – brackets in (HPLC) are not needed.

Lines 713-720 – the information here should be provided in past tense to match the style in other parts of the manuscript.

Line 725 – by should be replaced with from.

Lines 732-736 – the text should be revised to be grammatically correct.

Line 745 – the future tense is not suitable.

Line 749 – markers biomarkers should be checked.

This list is not complete. There are many more mistakes to be corrected in the whole manuscript.

CP showed a significant decline of sperm count (77%) and motility (50%) and life-dead% (42%), along with a 2-fold increase of sperm abnormalities compared to normal control rats.

SP contained 170 mg/ 100 g of iron, 160 mg/ 100 g of zinc and 20 mg/ 100 g of copper. It was reported that 273.197 mg/100 g of iron, 1.2154 mg/100g of copper, and 3.6229 mg/100g of zinc were detected in SP [19]. Other studies reported 87.4 mg/100 g of iron, 1.45 mg/100 g of zinc, and 0.47 mg/100 g of copper in SP [23] and these results were inconsistent with the results of this study as shown in Table S4.

Line 104: the tense was corrected to be “representing”.

This study focused on the phytochemical investigation of SP commercialized in the Egyptian markets as a raw powder sold in dietary stores and it is not manufactured as a pharmaceutical dosage form. The study includes analysis of amino acids, FAs (omega FAs), carotenes, vitamins, and minerals.

Author Response

Response Letter

Dear Editor,

Thank you for handling the manuscript ID: [molecules-1055207] entitled “Phytochemical and Biological Evaluation of a Newly Designed Nutraceutical Self Nano-Emulsifying Self Nano-Suspension for Protection and Treatment of Cisplatin Induced Testicular Toxicity in Male Rats”. We received the reviewers’ valuable comments and on behalf of all authors, I would like to express our sincere thanks to the reviewer who really raised some important points about this study. We have addressed the comments and revised the manuscript accordingly. Changes in the manuscript are highlighted.

Response to Reviewers Comments

Reviewer comments:

  1. Abstract is somewhat too long. According to the Instructions for Authors, it should be a maximum of 200 words.
  2. My suggestion is to include the information about the composition of the best self-nanoemulsifying self-nanosuspension suggested based on the research described in the manuscript in the Abstract. (blue highlight)

We have modified the abstract to include the detailed composition of the SNESNS. Additionally, we have reduced the number of abstract words to 198 words, Line (19-34)

“Abstract: The incorporation of cisplatin (CP) as a cytotoxic antineoplastic agent in most chemotherapeutic protocols is a challenge due to its toxic effect on testicular tissues. Natural compounds present a promising trend in research, so, a new nutraceutical formulation (NCF) was designed to diminish CP spermatotoxicity. A combination of three nutraceutical materials, 250 mg Spirulina platensis powder (SP), 25 mg Tribulus terrestris L. extract (TT), and 100 mg fish oil (FO) were formulated in self-nanoemulsifying self-nanosuspension (SNESNS). SP was loaded into the optimized self-nanoemulsifying system (30% FO, 50% span 80/cremophor EL and 20% isopropanol) and mixed with TT aqueous solution to form SNESNS. For the SP, phytochemical profiling revealed the presence of valuable amounts of fatty acids (FAs), amino acids, flavonoids, polyphenols, vitamins, and minerals. Transmission electron microscopy (TEM) and particle size analysis confirmed the formation of nanoemulsion-based nanosuspension upon dilution. Method validation of the phytochemical constituents in NCF has been developed. Furthermore, NCF was biologically evaluated on male Wistar rats and revealed the improvement of spermatozoa, histopathological features, and biochemical markers over the CP and each ingredient group. Our findings suggest the potential of NCF with SNESNS as a delivery system against CP-induced testicular toxicity in male rats.

  1. The same suggestion is even more important in the Conclusions part. What are the findings of the research carried out? Instead of repeating the same information from the Introduction, the emphasis on the results and the best of them should be included in Conclusions. The exact composition of the “newly designed formula of self-nanoemulsifying self-nanosuspension drug delivery system“ should be indicated.

We have modified the conclusion and the important points of the study are discussed and showed with the exact composition of the newly designed SNESNS drug delivery system at line (785-812)

Conclusion

“In conclusion, the current study provides evidence for the promising ameliorative effects of SNESNS on gonadotoxicity induced by CP. In the current study, the design of an NCF with SNESNS has been applied to mitigate or minimize CP-induced spermatotoxicy. It is the first time to design a unique formulation of three variable nutraceutical origin with different physical character in a homogenous stable SNESNS drug delivery system and each 1 g of the prepared formula composes of 250 mg SP which is blue-green algae, 100 mg FO which is of fish source and TT as a dried herb extract using Span 80/cremophor EL as surfactant and isopropanol as cosurfactant in the establishment of SNEDDS with FO in the percentage of (30% FO; 50% span80/cremophor EL; 20% isopropanol) and SP is loaded in the previous system alongside with TT aqueous solution to form a dark green homogenous, stable semisolid mixture with potential solubility and thus bioavailability. In the current study, The NCF of SNESNS design is used for the first time as an adjuvant supplement to diminish the toxic side effect of CP, especially in testicular tissues. Also, it is the first time to incorporate SP in pharmaceutical dosage form and a novel design (SNESNS) for treatment of spermatotoxicity. Analytical methods development and validation were employed to control the specific properties of the NCF. The same approach was applied for each ingredient comprised in the NCF. Strikingly, NCF showed the best near-to-normal histological and spermatogenic features comparing to the control group, suggesting the efficacy of NCF as a new therapeutic approach for the protection of the male reproductive system from the destructive toxic effects of CP. Additionally, TT, FO, and SP groups providing promising ameliorative effects of NCF on gonadotoxicity induced by CP. These beneficial actions were mediated, at least partly, via intervention with antioxidative, anti-inflammatory, and antiapoptotic events via disruption of Nrf2/NF-κB/caspase-3 cross-talk. The combination of the nutraceutical therapy with the newly designed formula of the SNESNS drug delivery system showed a synergistic effect with enhanced efficacy using poorly soluble ingredients. Further studies are necessary to clarify the exact molecular mechanisms for the NCF. Besides, future clinical studies are recommended to investigate the role of this new formulation as a dietary supplement involved in the chemotherapeutic protocol including CP as chemotherapy for young men with carcinoma.”

  1. The necessity of the Abbreviation section should be reconsidered. The two independent systems may be used: either abbreviation is provided in brackets after the full term when it is mentioned for the first time, or the abbreviation list is provided. Of course, with so many abbreviations used in the text, it might be easier for the reader to check the meaning of the abbreviation if needed. However, the expected readers of the article are professionals in the field, thus familiar with the terms.

We have removed the abbreviation list and used the abbreviation in brackets

 Maybe a table representing different SNESNSs with their composition indicated and main characterization parameters provided would add much-needed possibility to view the overall extent of the study and main results achieved.

The table for the composition of ternary diagram and characterization of SNEDDS is modified in the Supplementary file at Table S5

Table S5. Composition of ternary mixtures and Characterization of SNEDDSs

Formula code

% Fish oil

% Surfactant

(Span 80/Cremophore EL)

% Co-surfactant

(Isopropyl alcohol)

SNEDDS appearance

Emulsification Time (ET) ± SD

(s)

% Transmittance

± SD

Grade of formed Emulsion

F1

50

40

10

turbid liquid

100 ± 3.9

NA*

C

F2

50

30

20

turbid liquid

110 ± 4.5

NA

C

F3

50

20

30

turbid liquid

115 ± 3.4

NA

C

F4

40

40

20

turbid liquid

85 ± 4.5

NA

C

F5

40

30

30

turbid liquid

90 ± 3.5

NA

C

F6

40

20

40

turbid liquid

95 ± 4

NA

C

F7

35

60

5

clear liquid

36 ± 2.3

88.9 ± 2.6

B

F8

35

50

15

clear liquid

40 ± 2.1

87.8 ± 2

B

F9

35

40

25

clear liquid

45 ± 2.5

86.5 ± 2.4

B

F10

30

60

10

clear liquid

10 ± 1.2

91.5 ± 2.3

A

F11

30

50

20

clear liquid

12 ± 1.5

92.3 ± 1.6

A

F12

30

40

30

clear liquid

20 ± 1.5

89.7 ± 2.4

A

*NA means that the test was not done

  1. Author Contributions should be listed as described in the Instructions for Authors.

We have modified the author contribution as described in the Instructions for Authors at Line (835-839)

Author Contributions: "Conceptualization, S.R.A. and Z.T.A.; Phytochemical Experiment, S.R.A.; Development and characterization of SNEDDSs and SNESNS, D.M.N.A.; Histopathological Investigation, T.F.S.; Biochemical Investigation, E.R.; Validation, S.R.A.; Writing – Original Draft Preparation, S.R.A., D.M.N.A., T.F.S., and E.R.; Writing – Review & Editing, H.M.H., Z.T.A., and A.R.A.; All authors have read and agreed to the published version of the manuscript”.

  1. References should be described as requested by the journal –abbreviated journal names instead of full names and no issue number should be provided.

Action: The issue number is removed and the journal’s names are abbreviated

  1. Authors should check the manuscript critically to unify the terms (self nano-emulsifying self nano-suspension in the title, but self-nanoemulsifying self-nanosuspension or self nanoemulsifying in the manuscript; 50 –fold vs. 50 fold, etc.); regular and bald words „Table, Figure“ in the text. It is necessary to check for missing gaps between words and sentences starting from the lower-case letter. In general, a very careful revision is needed to reach a level of neat text, which would be suitable for publication.

Terms are unified to be in the form of “self-nanoemulsifying self-nanosuspension”

We correct “Table, Figure” to be bald and regular.

We add missing gaps between words and we carefully undergo revision and sometimes relocate some paragraphs to reach the paper to be the best.

  1. The English language of the manuscript requires a very careful revision as well. The manuscript cannot be read with pleasure. In some cases, one must force herself/himself to continue reading as the sentence structure is very cumbersome. The authors should revise the tenses, singular/plural nouns, and associated verbs. I would suggest writing the information, which has been published already and has already become a well-known truth, in the present perfect tense instead of past tense. Tense in the same paragraph should be checked to match.

The manuscript has been revised for the English language.

  1. Lines 108-110 – “It was noticed the presence of myristic, heptadecanoic, stearic, oleic, palmitoleic, omega-3, omega-6, linoleic acid (LA), GLA, and palmitic acid, the highest saturated fatty acid in SP, which was found to be in accordance with the results of this study [17].“ Maybe it could be replaced with „“Myristic, heptadecanoic, stearic, oleic, palmitoleic, omega-3, omega-6, linoleic acid (LA), GLA, and palmitic acid, the highest saturated fatty acid, were identified in SP. These results are in accordance with the ones in [17].“

Modified as suggested now at Line (110-112)

“There were 15 unsaponifiable matters (USM) (hydrocarbons) identified which represent 65.74% of the lipoidal content. Heptadecane was the main hydrocarbon identified (32.26% area) followed by squalene (8.1% area) as presented in Figure 1.A. while showed that there were 7 fatty acid methyl esters (FAME) were identified which represent 71.38% of the saponifiable matter. Hexadecanoic acid (palmitic acid) was the main fatty acid identified (25.53% area) followed by 9,12-octadecadienoic acid (linoleic acid) (15.08% area) and GLA (14.52% area) as presented in Figure 1.B. SP is considered an important vegetable source of omega fatty acids as GLA, linoleic and oleic acids [16]. Myristic, heptadecanoic, stearic, oleic, palmitoleic, omega-3, omega-6, linoleic acid (LA), GLA, and palmitic acid, the highest saturated fatty acid, have been identified in SP and these results are in accordance with the ones in [17].”

  1. Lines 133-134 – „It was the first time to estimate the TFC in commercialized SP in Egypt using rutin as a reference standard.“ Could be replaced with „ For the first time, TFC was estimated in commercial SP in Egypt using rutin as a reference standard.“

Modified as suggested now at Line (135-136)

“The TPC content was (58.64 mg gallic acid equivalent (GAE)/ g dry SP). It was reported that aqueous extract of SP exhibit the highest total phenolic content (43.2 mg GAE /g ext.) than 100% methanol extract (24.4 mg GAE /g ext.) [20] and (33.57 mg GAE/ g D. Wt.) [21] which were found to be slightly less than the results of this study. While, the TFC content was (0.864 mg rutin equivalent (RE)/ g dry SP). It was reported that SP contains total flavonoids (0.166 mg quercetin equivalent (QE)/g dry extract (DE) [22] and (7.11 mg QE/g DW) [18]. For the first time, TFC was estimated in commercial SP in Egypt using rutin as a reference standard.”

  1. Lines 136-140 – The sentences in the paragraphs like this “SP was found to contain (3.2 mg/100 g) cyanocobalamin (vitamin B12) and (191 mg/ 100g) ascorbic acid (vitamin C) while α-tocopherol (vitamin E) was undetectable. It was reported that the SP contains (175 μg vitamin B12 /100 g) and (9.86 mg vitamin E/100 g) [19]. Also, it was found that SP contains (162 138 μg/100 g) [23]. The SP was found to contain (60 mg vitamin E/100 g) [18] and these results were inconsistent with the results of this study as vitamin B12 was more than found in the previous studies and vitamin E was undetectable as shown in Table S3.“ could be corrected as follows: “SP contained 3.2 mg/100 g of cyanocobalamin (vitamin B12) and 191 mg/ 100g of ascorbic acid (vitamin C) while α-tocopherol (vitamin E) was undetectable. It was reported that 175 μg/100 g of vitamin B12 and 9.86 mg/100 g of vitamin E were detected in SP [19]. Other studies reported 138 μg/100 g of WHAT [23] and 60 mg/100 g vitamin E in SP [18]. However, these results were inconsistent with the results of this study as the content of vitamin B12 was higher than reported in the previous studies and vitamin E was undetectable as shown in Table S3.“

Modified as suggested now at Line (138-143)

“SP contained 3.2 mg/100 g of cyanocobalamin (vitamin B12) and 191 mg/ 100g of ascorbic acid (vitamin C) while α-tocopherol (vitamin E) was undetectable. It was reported that 175 μg/100 g of vitamin B12 and 9.86 mg/100 g of vitamin E were detected in SP [19]. Other studies reported 162 μg/100 g of vitamin B12 [23] and 60 mg/100 g vitamin E in SP [18] However, these results were inconsistent with the results of this study as the content of vitamin B12 was higher than reported in the previous studies and vitamin E was undetectable as shown in Table S3.”

Here are some more remarks:

  1. Line 19 – causes.

Rephrased the sentence as suggested now at Line (19-22)

“The incorporation of cisplatin (CP) as a cytotoxic antineoplastic agent in most chemotherapeutic protocols is a challenge due to its toxic effect on testicular tissues. Natural compounds present a promising trend in research, so, a novel nutraceutical formulation (NCF) was designed to diminish CP spermatotoxicity.”

  1. Line 26 – aside.

Rephrased the sentence as suggested now at Line (26-28)

For the SP, phytochemical profiling revealed the presence of valuable amounts of fatty acids (FA), amino acids, flavonoids, polyphenols, vitamins, and minerals.

  1. Line 41 – CP - Full name must be given when mentioned for the first time.

Corrected as suggested at Line 39

“Cisplatin (CP) is a main chemotherapeutic treatment that is used against different types of solid malignant tumors i.e. testicular germ cell tumor (TGCT)”

“Spirulina (SP)” full name is added at (Line 50)

“fatty acid (FAs)” full name is added at (Line 67)

“fish oil (FO)” full name is added at (Line 67)

“Self-Nanoemulsifying self-Nanosuspension (SNESNS)” full name is added at (Line 80)

  1. Lines 44-45 – the sentence should be corrected.

The sentence is corrected and rephrased as suggested at Line (42-44)

“Acute exposure to CP results in strong destruction of the seminiferous epithelium and this damage could be extended to the germ cell causing inhibition of its proliferation, meiosis, and differentiation to spermatozoa.”

  1. Line 48 and elsewhere– my suggestion would be to replace the word “embarrassment” with “retardation” or “obstruction” as the word embarrassment has also the other meaning.

The sentence is corrected and rephrased as suggested at Line (47)

“peroxidation resulting in protein synthesis retardation and DNA destruction [3]”

  1. Line 68 - linoleic acid is the main FA in FO.

The sentence is corrected and rephrased as suggested at Line (66-67)

“Eicosapentaenoic acid (EPA), docosahexaenoic acid (DHA) and alpha-linolenic acid are the main FAs in fish oil (FO)”

  1. Lines 155-156 – the sentence should be corrected.

The sentence is corrected and rephrased as suggested at Line (160-161)

It showed the highest solubility in span 80/cremophor EL as a surfactant and isopropyl alcohol as a co-surfactant.

  1. Line 190 – eliminated instead of omitted.

The sentence is corrected at line 196

  1. Line 219 – starting with this place abbreviation SP has two meanings, what is not acceptable - suspended drug (SP)

The sentence is corrected as suggested at Line (225-226)

“Besides, stirring SNESNS in distilled water resulted in spontaneous nano-sizing of the suspended SP in SNESNS and the formation of globular nanoparticles as shown in Figure 5. A and B “

  1. Lines 264-266 – how can the content be calculated by HPLC?

The sentence is corrected and rephrased as suggested at Line (272-275)

The calculated content of β-carotene in SP and NCF by HPLC is (0.44 mg/g pd. and 0.17 mg/g formula), respectively. The calculated content of β-carotene in SP and NCF by HPTLC is (1.63 mg/g pd.) for SP, while β-carotene was not separated from the NCF by HPTLC, these results were shown in Figure 7 & 8.

The content is calculated as:

We perform a standard calibration curve of the β-carotene standard at 4 different concentrations giving the equation “y = 54020x – 9.6388”, R2 = 0.9995 and a slope equals 54020. Then for SP, we took 1 g of SP powder to be finally concentrated as (20 mg SP/ml), the area under peak (AUP) for SP by HPLC was (468.242) and this reading was compensated in the previous equation to give (0.008846 mg β-carotene /ml) then the following equation is used:

β-carotene (%w/w) =

C: Sample’s β-carotene concentration from the calibration curve (0.008846 mg/ml).

FV: The final volume of the sample preparation (50 ml)

W: The sample weight (1000 mg).

% of β-carotene in SP = 0.04423% and this equal 0.44 mg β-carotene/ g of powder

For NCF:

We took 1.18 g from the new formula (equivalent to 295 mg SP) to be finally concentrated as (5.9 mg SP/ml), the area under peak (AUP) for NCF by HPLC was (177.236) and this reading was compensated in the previous equation to give (0.003459 mg β-carotene /ml) then the following equation is used:

β-carotene (%w/w) =

C: Sample’s β-carotene concentration from the calibration curve (0.003459 mg/ml).

FV: The final volume of the sample preparation (50 ml)

W: The sample weight (295 mg).

% of β-carotene in NCF = 0.058% and this equal 0.17 mg β-carotene/ g formula

  1. Line 275 – were injected.

The sentence is corrected as suggested at Line (285)

“each analyte were injected in triplicate.”

  1. Lines 303-305 – the sentence should be corrected to communicate the meaning.

The sentence is corrected and rephrased as suggested at Line (313-317)

“The precision (repeatability) was performed for UV estimation of the total steroidal content method in two manners; the intra-day and inter-day precision expressed as %RSD which is found to be less than 2% for intra-day precision and more than 2% for inter-day precision. It is important to take into consideration that the samples should be freshly prepared before being measured on a UV spectrophotometer and not to be stored even in a refrigerator.”

  1. Table 2 – gm should be replaced with g.

We corrected “gm” and replaced it with “g” in Table 2, Line 324

“Each 1 g new formula”

  1. Line 348 – Tables.

We corrected “Table” and replaced it with “Tables” at the line (362-363)

  1. Line 421-424 – the sentence should be corrected; the tenses should be matched.

The sentence is corrected and rephrased as suggested at Line (435-438)

“CP is a strong alkylating chemotherapeutic agent and its consumption in many chemotherapeutic protocols can lead to harmful side effects including testicular toxicity through induction the release of reactive oxygen species (ROS) that causes disequilibrium which is occurred between the synthesis of oxidants and removal by antioxidants defense mechanism”

  1. Materials and methods section – the word “supplied from” should be replaced with “purchased” or “obtained”. “Supplied by” could be used indicating that some other person or company gave something to the authors and in line 538.

We corrected the word “supplied” and replaced it with “purchased” as suggested at lines (540-562)

  1. Line 592 – “Half gram“ should be written as 0.5 g.

“Half gram” is replaced with 0.5 g at Line 605

  1. Line 602 – was instead of is – tenses should match.

Corrected as suggested “hazy solution was reached” at line 615

  1. Lines 657-658 – this information is not suitable in the Materials and Methods section, since the exact procedures must be provided here. All discussions should be moved to the Discussion section.

The required correction was done. Part of this section was moved to Discussion and only the method of performing the test was kept in this section

  1. Lines 677 and 680 – my suggestion would be to write just „HPLC investigation“ and „HPTLC investigation“.

We corrected the sentences as suggested “HPLC investigation” and “HPTLC investigation”, Line (693-696)

  1. Line 679 – brackets in (HPLC) are not needed.

We removed the brackets and become “HPLC”; Line 694

  1. Lines 713-720 – the information here should be provided in the past tense to match the style in other parts of the manuscript.

The tense of the words is changed to past tense as suggested at Line (729-736)

“Sixty rats were allocated randomly into six groups. The first group represented the normal control group (NC) that was administrated saline orally. Testicular toxicity was induced by a single intraperitoneal (IP) dose of CP (7 mg/kg) [80] on day 11th. Rats were treated for 20 consecutive days, 10 days before, and 10 days after the administration of CP. The second group included the untreated rats that were administrated CP only (NC group). In groups 3, 4, 5, and 6, rats treated orally with TT at a dose of (100 mg/kg/day) [81]; SP at a dose of (1000 mg/kg/day) [82]; FO at a dose of (400 mg/kg/day) and NCF group at a dose of (4.4 g /kg/day) . The last dose of any treatment was given 24 h before the end of the experiment.”

  1. Line 725 – by should be replaced with from.

The word “by” was replaced with “from” at Line 741

  1. Lines 732-736 – the text should be revised to be grammatically correct.

We corrected the paragraph grammatically as suggested at Line (748-754)

“The in-vitro antioxidant activity for SP and NCF was performed according to [85]. The DPPH was prepared at a concentration of 0.01 % in ethanol (HPLC grade). The SP was dissolved in absolute ethanol and at different concentrations of 0.18, 0.36, 0.54, 0.72, and 0.9 mg/ml, and equivalent concentrations from NCF were prepared as mentioned above then 1 ml of each test solution was added to 6 ml of DPPH solution. A blank sample was run using absolute ethanol. The test solutions were incubated for 30 min. at room temperature and the absorbance was recorded against the blank at 517 nm.”

  1. Line 745 – the future tense is not suitable.

The tense of the sentence is corrected as suggested to be “Dead animals were frozen till their incineration” at the line (762-763)

  1. Line 749 – markers biomarkers should be checked.

Corrected to be “markers” and “biomarkers” removed at line 766

Other corrections not mentioned by reviewers:

  • “CP showed a significant decline of sperm count (77%) and motility (50%) and life-dead% (42%), along with a 2-fold increase of sperm abnormalities compared to normal control rats.” Was transferred from “Effect of cisplatin-induced alterations in serum testosterone” to “Sperm count and motility” in Result, Line (328-330).
  • “SP contained 170 mg/ 100 g of iron, 160 mg/ 100 g of zinc and 20 mg/ 100 g of copper. It was reported that 273.197 mg/100 g of iron, 1.2154 mg/100g of copper, and 3.6229 mg/100g of zinc were detected in SP [19]. Other studies reported 87.4 mg/100 g of iron, 1.45 mg/100 g of zinc, and 0.47 mg/100 g of copper in SP [23] and these results were inconsistent with the results of this study as shown in Table S4. Modified and rephrased at the line (145-149)
  • “The total protein content represented77% of SP, which was found to be in accordance with the labeled amount of the total protein (65-75% protein content) in the marketed product. The essential amino acids represented 28.7% while non-essential amino acids represented 40% from SP. Glutamic acid is the main non-essential amino acid (14.1%) followed by aspartic acid (8.52 %) while leucine is the main essential amino acid represent (6.4%) followed by valine (5.3%) as presented in Figure 2. Previously, the total protein content was reported to be 56.79% [18] and other studies reported it to be 62.84% [19], which was slightly below the results of this study. Other studies reported leucine as the main essential amino acid with a percentage of 5.5% and glutamic acid as the main non-essential amino acid with a percentage of 9.2% [17], which was in accordance with the results of this study“ Tenses corrected and modified at the line (117-125)
  • Line 104: the tense was corrected to be “representing”
  • “This study focused on the phytochemical investigation of SP commercialized in the Egyptian markets as a raw powder sold in dietary stores and it is not manufactured as a pharmaceutical dosage form. The study includes analysis of amino acids, FAs (omega FAs), carotenes, vitamins, and minerals.” Corrected for tenses, the abbreviation was added and modified at the line (99-102)
  1.  

Reviewer 2 Report

The manuscript reported the incorporation of a combination of three herbal materials, Spirulina platensis powder (SP), Tribulus terrestris L. extract(TT) and fish oil (FO) in the toxicity assay. The results showed that the protective effect of the SNESNS against induced testicular toxicity on male wistar rats. However, the mechanistic actions of the SP needs to be elaborated. How can the toxicity be reduced? Furthermore, the results that suggest the potential of the novel NCF with SNESNS as a
delivery system against CP induced testicular toxicity in male rats need to be substantiated with supportive data. It is more useful
further test clinically as an adjuvant therapy with chemotherapy can be demonstrated.

Author Response

Reviewer comments #2

  1. The mechanistic actions of the SP need to be elaborated

The mechanistic action of SP was clarified at Line (457-464), Line (477-486), and Line (490-492).

  1. How can the toxicity be reduced?

SP is reported to be safe and clarified at (line 52)

  1. It is more useful for further test clinically as an adjuvant therapy with chemotherapy can be demonstrated.

This study was performed as a preliminary study only to evaluate the effect of the newly designed nutraceutical formulation in experimental animals and the results found in this study, of course, was recommended to perform further clinical studies, the clinical trials in Egypt have very restricted regulations and not performed till the product is registered in the Egyptian Ministry of Health which also subject to very routine laws and regulations this study need great fund as “Project” in collaboration with the ministry of health.